# Quantitative mapping of human hair greying and reversal in relation to life stress

Ayelet M Rosenberg[1], Shannon Rausser[1], Junting Ren[2], Eugene V Mosharov[3,4], Gabriel Sturm[1], R Todd Ogden[2], Purvi Patel[5], Rajesh Kumar Soni[5], Clay Lacefield[4], Desmond J Tobin[6], Ralf Paus[7,8,9], Martin Picard[1,4,10]*

[1]Department of Psychiatry, Division of Behavioral Medicine, Columbia University Irving Medical Center, New York, United States; [2]Department of Biostatistics, Mailman School of Public Health, Columbia University Irving Medical Center, New York, United States; [3]Department of Psychiatry, Division of Molecular Therapeutics, Columbia University Irving Medical Center, New York, United States; [4]New York State Psychiatric Institute, New York, United States; [5]Proteomics and Macromolecular Crystallography Shared Resource, Columbia University Irving Medical Center, New York, United States; [6]UCD Charles Institute of Dermatology & UCD Conway Institute, School of Medicine, University College Dublin, Dublin, Ireland; [7]Dr. Phillip Frost Department of Dermatology & Cutaneous Surgery, University of Miami Miller School of Medicine, Miami, United States; [8]Centre for Dermatology Research, University of Manchester, Manchester, United Kingdom; [9]Monasterium Laboratory, Münster, Germany; [10]Department of Neurology, H. Houston Merritt Center, Columbia Translational Neuroscience Initiative, Columbia University Irving Medical Center, New York, United States

*For correspondence:
martin.picard@columbia.edu

Competing interests: The authors declare that no competing interests exist.

## Abstract

**Background:** Hair greying is a hallmark of aging generally believed to be irreversible and linked to psychological stress.

**Methods:** Here, we develop an approach to profile hair pigmentation patterns (HPPs) along individual human hair shafts, producing quantifiable physical timescales of rapid greying transitions.

**Results:** Using this method, we show white/grey hairs that naturally regain pigmentation across sex, ethnicities, ages, and body regions, thereby quantitatively defining the reversibility of greying in humans. Molecularly, grey hairs upregulate proteins related to energy metabolism, mitochondria, and antioxidant defenses. Combining HPP profiling and proteomics on single hairs, we also report hair greying and reversal that can occur in parallel with psychological stressors. To generalize these observations, we develop a computational simulation, which suggests a threshold-based mechanism for the temporary reversibility of greying.

**Conclusions:** Overall, this new method to quantitatively map recent life history in HPPs provides an opportunity to longitudinally examine the influence of recent life exposures on human biology.

**Funding:** This work was supported by the Wharton Fund and NIH grants GM119793, MH119336, and AG066828 (MP).

## Introduction

Hair greying is a ubiquitous, visible, and early feature of human biological aging (*O'Sullivan et al., 2021*; *Tobin, 2011*). The time of onset of hair greying varies between individuals, as well as between

**eLife digest** Hair greying is a visible sign of aging that affects everyone. The loss of hair color is due to the loss of melanin, a pigment found in the skin, eyes and hair. Research in mice suggests stress may accelerate hair greying, but there is no definitive research on this in humans. This is because there are no research tools to precisely map stress and hair color over time. But, just like tree rings hold information about past decades, and rocks hold information about past centuries, hairs hold information about past months and years.

Hair growth is an active process that happens under the skin inside hair follicles. It demands lots of energy, supplied by structures inside cells called mitochondria. While hairs are growing, cells receive chemical and electrical signals from inside the body, including stress hormones. It is possible that these exposures change proteins and other molecules laid down in the growing hair shaft. As the hair grows out of the scalp, it hardens, preserving these molecules into a stable form. This preservation is visible as patterns of pigmentation. Examining single-hairs and matching the patterns to life events could allow researchers to look back in time through a person's biological history.

Rosenberg et al. report a new way to digitize and measure small changes in color along single human hairs. This method revealed that some white hairs naturally regain their color, something that had not been reported in a cohort of healthy individuals before. Aligning the hair pigmentation patterns with recent reports of stress in the hair donors' lives showed striking associations. When one donor reported an increase in stress, a hair lost its pigment. When the donor reported a reduction in stress, the same hair regained its pigment. Rosenberg et al. mapped hundreds of proteins inside the hairs to show that white hairs contained more proteins linked to mitochondria and energy use. This suggests that metabolism and mitochondria may play a role in hair greying. To explore these observations in more detail Rosenberg et al. developed a mathematical model that simulates the greying of a whole head of hair over a lifetime, an experiment impossible to do with living people. The model suggested that there might be a threshold for temporary greying; if hairs are about to go grey anyway, a stressful event might trigger that change earlier. And when the stressful event ends, if a hair is just above the threshold, then it could revert back to dark.

The new method for measuring small changes in hair coloring opens up the possibility of using hair pigmentation patterns like tree rings. This could track the influence of past life events on human biology. In the future, monitoring hair pigmentation patterns could provide a way to trace the effectiveness of treatments aimed at reducing stress or slowing the aging process. Understanding how 'old' white hairs regain their 'young' pigmented state could also reveal new information about the malleability of human aging more generally.

---

individual hair follicles, based on genetic and other biobehavioral factors (*Akin Belli et al., 2016*; *Bernard, 2012*). But most people experience depigmentation of a progressively large number of hair shafts (HSs) from their third decade onward, known as achromotrichia or canities (*Panhard et al., 2012*). The color in pigmented HSs is provided by melanin granules, a mature form of melanosomes continuously supplied to the trichocytes of the growing hair shaft by melanocytes of the hair follicle pigmentary unit (HFPU) (*Tobin, 2011*). Age-related greying is thought to involve bulb and outer root sheath melanocyte stem cell (MSC) exhaustion (*Commo et al., 2004*; *Nishimura et al., 2005*), neuroendocrine alterations (*Paus, 2011*), and other factors, with oxidative damage to the HFPU likely being the dominant, initial driver (*Arck et al., 2006*; *Paus, 2011*; *Trueb and Tobin, 2010*). While loss of pigmentation is the most visible change among greying hairs, depigmented hairs also differ in other ways from their pigmented counterparts (*Tobin, 2015*), including in their growth rates (*Nagl, 1995*), HF cycle, and other biophysical properties (*Van Neste and Tobin, 2004*). Hair growth is an energetically demanding process (*Flores et al., 2017*) relying on aerobic metabolism in the HF (*Williams et al., 1993*), and melanosome maturation also involves the central organelle of energy metabolism, mitochondria (*Basrur et al., 2003*; *Zhang et al., 2019*). Moreover, mitochondria likely contribute to oxidative stress within the HF (*Lemasters et al., 2017*), providing converging evidence that white hairs may exhibit specific alterations in mitochondrial energy metabolism.

Although hair greying is generally considered a progressive and irreversible age-related process, with the exclusion of alopecia areata (*McBride and Bergfeld, 1990*), cases of drug- and mineral deficiency-induced depigmentation or repigmentation of hair have been reported (*Kavak et al., 2005*; *Kobayashi et al., 2014*; *Komagamine et al., 2013*; *Reynolds et al., 1989*; *Ricci et al., 2016*; *Sieve, 1941*; *Yoon et al., 2003*) reflecting the influence of environmental inputs into HFPU function (*Paus et al., 2014*). Because most hairs are continually growing from a living hair follicle, sensitive to changing physiological conditions, into a hardened hair shaft external to the body that retains stable molecular traces of these conditions, the hair shaft represents a bioarchive of recent exposures (*Kalliokoski et al., 2019*). While spontaneous repigmentation can be pharmacologically induced, its natural occurrence in unmedicated individuals is rare and has only been reported in a few single-patient case studies (*Comaish, 1972*; *Navarini and Trüeb, 2010*; *O'Sullivan et al., 2021*; *Tobin and Cargnello, 1993*; *Tobin and Paus, 2001*). The reversal of hair greying has not been quantitatively examined in a cohort of healthy adults, in parallel with molecular factors and psychosocial exposures.

The influence of psychological stress on hair pigmentation is a debated, but poorly documented, aspect of hair greying. In humans, psychological stress accelerates biological aging as measured by telomere length (*Epel et al., 2004*; *Puterman et al., 2016*). In mice, psychological stress and the stress mediator norepinephrine acutely causes depigmentation (*Zhang et al., 2020*). However, greying in both mice and humans has been shown to be a relatively irreversible phenomenon driven in part by a depletion of melanocyte stem cells, although some stem cells and transient amplifying cells do remain (*Trueb and Tobin, 2010*). In humans, recent evidence suggests hair growth and pigmentation changes in response to stress (*Peters et al., 2017*), but this relationship, along with reversal of greying, remain insufficiently understood. The paucity of quantitative data in humans is mostly due to the lack of sensitive methods to precisely correlate stressful psychobiological processes with hair pigmentation and greying events at the single-follicle level.

Here, we describe a digitization approach to map hair pigmentation patterns (HPPs) in single hairs undergoing greying and reversal transitions, examine proteomic features of depigmented white hairs, and illustrate the utility of the HPP approach to interrogate the association of life stress and hair greying in humans. Because previous literature suggests that rare repigmentation events are more likely to occur in the early stages of canities (*Van Neste and Tobin, 2004*), the current study focuses primarily on pigmentation events in young to middle-aged participants. Finally, we develop a computational model of hair greying to explore the potential mechanistic basis for stress-induced greying and reversibility on the human scalp hair population, which could potentially serve as a resource for the in silico modeling of macroscopic aging events in human tissues.

## Materials and methods

### Participants

The study was approved by New York State Psychiatric Institute (NYSPI IRB Protocol #7748). All participants provided written informed consent for their participation in this study and to the publications of data. Dark, white, and bi-color hairs were collected from healthy participants self-identified as 'having some grey hairs' or 'two-colored hairs'. Exclusion criteria included the use of dye, bleaching, or other chemical treatments on hairs. In addition, participants with hairs shorter than approximately 4 cm were excluded. Participants were recruited via local advertisement and using a snowball recruitment strategy. Some participants were staff of NYSPI and Columbia University Irving Medical Center, but no patients participated in the study. Eligible participants were asked to provide all two-colored hairs present on their scalps or other body regions. A total of 14 individuals (seven females, seven males), mean age 35 ± 13 (SD, range: 9–65), were recruited. Hairs were plucked, manually or with standard flat tip tweezers, from the scalp or other body regions and archived for future imaging or molecular analyses. Only plucked hairs with follicular tissue attached (excluding broken hairs) were used in analyses to enable the interpretation of the direction of pigmentation – transition if the HS tip is dark and the root white, reversal if the HS tip is white and the root dark. Where possible, participants with two-colored hairs also provided fully dark or white hairs for comparison.

While the participant age range does not capture the typical range of an aged population, it provides an opportune window to examine the beginning of the aging process as it corresponds to the

typical age of onset for greying or canities (*Tobin, 2009*). The hair follicles also manifest stochastic hair-to-hair heterogeneity similar to that seen between individual cells in aging organs (*Bernard, 2012*; *van Deursen, 2014*). For this reason, likely as a result of stochastic processes similar to those that drive cellular heterogeneity in gene expression, some HFs reach the end of their pigmented life even in relatively young individuals. Although the course and process of aging in middle age may differ from later aging, rare regimentation events are more likely to occur in the early stages of canities (O'Sullivan, 2020; *Van Neste and Tobin, 2004*). This is in accordance with our mathematical model (Figure 5) and with the isolated cases of repigmentation reported in the literature. Thus, to capture this phenomenon without additional confounds that could arise from systemic aging (comorbidities, systemic inflammation, or other), we focused our investigation on this particular age window. As our model predicts, this age window also increases the probability that hairs are near their greying threshold, and as such have the possibility to undergo observable reversal.

## Hair imaging

Whole hairs were first photographed using a Panasonic DC-FZ80 Digital Camera against a white background, with the hair tip and follicle systematically oriented. To facilitate visualization of the images of whole hairs in the figures (photographic insets of whole hairs), the exposure, saturation, sharpness and light/dark tones of the photographs were enhanced. For microscopic imaging of hair follicles and HFPU, individual hair shafts and root-ends were imaged with an Olympus BX61 upright microscope (Olympus BX61 Upright Wide Field Microscope, RRID:SCR_020343) equipped with a digitized stage (ProScan; Prior Scientific), a 2.5x/0.075 air (Zeiss, Germany), 10x/0.4 air (Zeiss, Germany) or 40x/1.3 oil (Olympus, MA) objectives, using DP71 camera (Olympus, MA) and MetaMorph software (MetaMorph Microscopy Automation and Image Analysis Software, RRID:SCR_002368) (Molecular Devices, CA) version 7.7.7.0. Images were scaled and analyzed in ImageJ (ImageJ, RRID: SCR_003070) (version 1.42q, NIH, http://rsb.info.nih.gov/ij). For microscopic imaging of hair shafts and videos of HPP transitions along the length of hair shafts, hairs were temporarily mounted with water on a glass slide (10x magnification, 15 ms exposure, 24-bit, ISO 1600, 4080 × 3072 digitizer).

## Digitization of HPPs

To generate high-resolution HPPs, HSs were digitized as high-resolution 8-bit Greyscale images (3200dpi, default adjustments, Epson Perfection V800 Photo Scanner), and the scanned images were processed using Image J (Fiji, RRID:SCR_002285). To capture both the white and dark sections of each hair, hairs were immobilized onto a smooth surface by taping the plucked hair follicle (proximal to the epidermis), straightening the entire length as much as possible without placing too much force on the hair, and immobilizing the tip (distal to epidermis) with adhesive tape. HSs were dry and were checked for potential knots or twists caused by handling. Any dust was removed from the hair surface using a kimwipe before being placed on the scanner. Areas of each hair between the immobilized ends were used for analyses. To extract hair darkness at each point along the length, pixel luminosity at each position was estimated as the darkest value across a sliding one-dimensional pixels array perpendicular to the shaft axis, where the hair itself represented the darkest area, and HPP graphs were generated by plotting the intensity in arbitrary units (A.U.) by distance (cm). Intensity ranged from 0 to 255 A.U., with 0 being white and 255 being black. The data was then denoised using a 100-pixels rolling average, and the resulting HPP was imported into Prism 8 (GraphPad Prism, RRID:SCR_002798) for visualization.

To compare intensity profiles across multiple hairs, we transformed numerical intensity values by normalizing to the average intensity of each hair. A total of 100 randomly selected dark hairs were manually plucked from one female and one male individual, including 25 hairs per head region (left and right temporal, top, and crown). Digitized hairs for each individual were graphed as a heatmap, grouped by head region. To examine the hypothesis that hairs exhibit regional variation in HPPs, the intensity of all 25 hairs per region were then averaged to create an 'average' hair from each region. A plot was then made for the four 'average hairs', one from each head region (*Figure 5—figure supplement 1*).

## Electron microscopy

Dark and white scalp hairs were plucked from two healthy individuals: a 38-year-old African-American male and a 33-year-old Caucasian male. The African-American hairs were curly and black, while the Caucasian hairs were straight and auburn. Hairs were fixed in a 2% glutaraldehyde solution in 0.1 M cacodylate (TAAB Lab Equipment) buffer, pH 7.4 as described previously (*Picard et al., 2013*). Briefly, plucked hair shafts were cut to 2–3 cm in length, immersed in fixative, and incubated at room temperature for 2 weeks. HS were then post-fixed and dehydrated in ethanol, cut into smaller segments of 0.5 cm, and embedded in longitudinal orientation in 100% resin. Orientation and section quality were confirmed with 1 μm-thick sections, and ultrathin sections of 70 nm were cut using a diamond knife on a Leica EM UC7 ultramicrotome (Leica EM UC7 ultramicrotome, RRID:SCR_016694). Sections mounted on Pioloform filmed copper grids prior to staining with 2% aqueous uranyl acetate and lead citrate (Leica). Ultrathin sections were examined on a Phillips CM 100 Compustage (FEI) transmission electron microscope and digital micrographs were captured by an AMT CCD camera.

Matched dark and white hairs from the donors were imaged, and three different segments along each hair were analyzed. All images used for analysis were captured at ×7500 magnification, with a pixel size of 0.00902 μm/pixel. Images were imported into ImageJ for analysis and all melanin granules contained within a given picture were manually traced (Intuos tablet). In each photograph, the intensity of the melanin granules, cortex, and background (outside the hair) were quantified. Cortex and melanin granule intensity were normalized by subtracting the background average intensity (measured from three different standard regions of interest – ROIs) to ensure comparability of various micrographs, hair segments, and between dark and white hairs. The intensity of the cortex was also quantified from eight different ROIs devoid of melanin granules.

To compute melanin granule size, we obtained the surface area of each manually traced granule. To compute melanin granule density per hair region, the total cortex area in each scaled micrograph was recorded and was divided by the total number of granules that were found in that image, yielding the number of granules/μm$^2$, which was then multiplied by 100.

## Hair shaft proteomics

The protocol in both experiments 1 and 2 for hair digestion were adapted from a previous protocol establishing that SDS-based protein extraction methods result in higher protein yield than urea-based digestion (*Adav et al., 2018*). This method was adapted with the addition of an initial mechanical homogenization step to extract proteins from minimal amounts of hair tissue (1–2 cm), which was necessary to analyze multiple hair segments along the same HS. After incubation, the hair was reduced with DTT and then alkylated with iodoacetamide (IAA), as per previous methods (*Goecker et al., 2020*).

### Experiment 1

For label-free quantitative proteomics, a 2 cm segment of plucked dark and white HS matched for distance relative to the follicle end was isolated from one female and one male participant. Each HS was washed independently in 20% methanol, ground and extracted in a glass homogenizer with SDS in Tris-buffered saline with 150–200 μl of 4% protease inhibitor cocktail (Sigma P8300), precipitated with chloroform-methanol, redissolved in 8 M urea with ammonium bicarbonate, reduced, alkylated and digested with trypsin. For liquid chromatography and mass spectrometry, two technical replicas (161 min chromatograms) were recorded for each sample. Separations were performed with an Ultimate 3000 RSLCNano (Thermo Scientific) on a 75 μm ID x 50 cm Acclaim PepMap reversed phase C18, 2 μm particle size column. Chromatographic flow rate was 300 nL/min with an acetonitrile/formic acid gradient. The liquid chromatograph was coupled to a Q Exactive HF mass spectrometer (Thermo QE-HFX mass spectrometer, RRID:SCR_018703) (Thermo Scientific) using data-dependent acquisition. Data were searched against a Swiss-Prot human protein database with Mascot v.2.5.1 (Matrix Science Ltd., RRID:SCR_000307). Semi-quantitative exponentially-modified protein abundance index (emPAI) was calculated by the Mascot software. A total of 744 proteins were detected. Proteins not detected in two or more samples from a total of eight were not included for further analyses. Among the eligible proteins (n = 323), the fold change in protein abundance was

compared between white and dark hairs. The gene list (*Supplementary file 1*) used for downstream analyses includes downregulated (<0.8 fold, n = 23) and upregulated (>1.5 fold, n = 67) proteins.

## Experiment 2

One cm hair samples from each subject (n = 17) were washed in 1 ml of 20% methanol while agitating at 1400 rpm for 20 mins at room temperature. The washed hair samples were homogenized using 150 µL of lysis buffer (4% SDS/0.1 M Tris/Protease inhibitor cocktail) in a glass homogenizer until no hair particles are visible. The lysates were incubated at 65°C for 13 hr overnight at 1500 rpm. The next day, the samples were centrifuged at 20,000 x g for 10 min to clear the lysate. Cleared lysates were reduced with 5 mM DTT at room temperature for 30 min at 1000 rpm. Alkylation was carried out with 11 mM IAA at room temperature in the dark for 30 min and quenched with 5 mM DTT for 15 min at room temperature. The proteins were precipitated using a chloroform-methanol method, and precipitated protein pellets were dissolved in 15 µL of resuspension buffer (4 M Urea/ 0.1 M Tris) and sonicated until entirely homogenized. Protein concentration was estimated using the BCA assay. 4 µg of total proteins from each sample was digested for 4 hr at 37°C with Lys-C protease at a 50:1 protein-to-protease ratio while shaking. Samples were then diluted with 100 mM Tris to bring down the urea concentration to the final 1.6 M and digested further with trypsin was then added at a 100:1 protein-to protease ratio, and the reaction was incubated overnight at 37°C. The next day, digestion was stopped by the addition of 1% TFA (final v/v) and centrifuged at 14,000 g for 10 min at room temperature. Cleared digested peptides were desalted on SDB-RP Stage-Tip and dried in a speed-vac. Peptides were dissolved in 3% acetonitrile/0.1% formic acid and 200 ng of peptides were injected on an Orbitrap Fusion Tribrid mass spectrometer (Thermo Orbitrap Fusion Tribrid Mass Spectrometer, RRID:SCR_020559) (Thermo Scientific) coupled to an UltiMate 3000 UHPLC (Thermo Scientific Dionex Ultimate 3000 system, RRID:SCR_020563) (Thermo Scientific). Peptides were separated on a 25 cm column (i.d. 75 µm, EASY-Spray, 2 µm, 100 Å) using a non-linear gradient of 5–35% at a flow rate of 300 nL/min using a buffer B (0.1% (v/v) formic acid, 100% acetonitrile) for 90 min. After each gradient, the column was washed with 90% buffer B for 5 min and re-equilibrated with 98% buffer A (0.1% formic acid, 100% HPLC-grade water) for 15 min. The full MS spectra were acquired in the Orbitrap from 400 to 1500 m/z at 120K with a $2 \times 10^5$ ion count target and a maximum injection time of 50 ms. The instrument was set to run in top speed mode with 3 s cycles for the survey and the MS/MS scans. After a survey scan, MS/MS was performed on the most abundant precursors, that is, those exhibiting a charge state from 2 to 6 of greater than $5 \times 10^3$ intensity, by isolating them in the quadrupole at 1.6 Th. We used collision-induced dissociation (CID) with 35% collision energy and detected the resulting fragments with the rapid scan rate in the ion trap. The automatic gain control (AGC) target for MS/MS was set to $1 \times 10^4$, and the maximum injection time was limited to 35 ms. The dynamic exclusion was set to 30 s with a 10-ppm mass tolerance around the precursor and its isotopes. Monoisotopic precursor selection was enabled.

Raw mass spectrometric data were analyzed using the MaxQuant environment v.1.6.1.0 (MaxQuant, RRID:SCR_014485) (*Thompson et al., 2008*) and Andromeda (*Cox et al., 2011*) for database searches at default settings with a few modifications. The default is used for first search tolerance and main search tolerance (20 ppm and six ppm, respectively). MaxQuant was set up to search with the reference human UniProtKB proteome database downloaded from UniProt (UniProtKB, RRID: SCR_004426). MaxQuant performed the search trypsin digestion with up to two missed cleavages. Peptide, site, and protein false discovery rates (FDR) were all set to 1% with a minimum of 1 peptide needed for identification; label-free quantitation (LFQ) was performed with a minimum ratio count of 1. To discriminate between relative and absolute protein abundance, we also examined the intensity-based absolute quantification (iBAQ) values for each protein (*Schaab et al., 2012*). This allowed to discriminate between relative versus absolute upregulation of mitochondrial proteins. In particular, analysis of iBAQ data ensured that highly stoichiometric keratins and keratin-associated proteins were not downregulated in white hairs, and thus could not account for the observed upregulation of mitochondrial and other proteins. Search criteria included carbamidomethylation of cysteine as a fixed modification, oxidation of methionine, acetyl (protein N-terminus) and deamination for asparagine or glutamine (NQ) as variable modifications. A total of 438 proteins were detected. Proteins not detected in at least three samples from a total of 11 were not included in downstream analyses. Detected proteins not mapped to known genes are labeled as 'unknown'. From the eligible proteins

(n = 192), the fold change in protein abundance was compared between white and dark hairs. The gene list (*Supplementary file 2*) used for downstream analyses includes downregulated (<0.8 fold, n = 56) and upregulated (>1.5 fold, n = 106) proteins. In sensitivity analyses, the data was re-analyzed with an alternate criterion of detection in at least two samples per color ($\geq$2/6 dark,$\geq$2/5 white) and the data presented in *Figure 1—figure supplement 1*.

For both Experiment 1 and 2, functional enrichment analysis was performed at an FDR threshold of 0.05 using Gene Ontology (GO) (Gene Ontology, RRID:SCR_002811) (http://geneontology.org/) and Kyoto Encyclopedia of Genes and Genomes (KEGG) (Kyoto Encyclopedia of Genes and Genomes Expression Database, RRID:SCR_001120) (https://www.genome.jp/kegg/kegg1.html) annotations in ShinyGO v.0.61 (ShinyGO, RRID:SCR_019213) (http://bioinformatics.sdstate.edu/go/) (*Ge et al., 2019*). Protein-protein interaction (PPI) networks were generated, analyzed for network metrics, and visualized in STRING v.11.0 (STRING, RRID:SCR_005223) (https://string-db.org/cgi/input.pl) (*Szklarczyk et al., 2019*). Given the substantial representation of mitochondrial proteins among upregulated lists (67 of 323, 26.8% in Experiment 1; 21 of 106, 19.8% in white vs dark Experiment 2), we queried MitoCarta 2.0 (MitoCarta, RRID:SCR_018165) (*Calvo et al., 2016*) and other sources, including GeneCards.org (GeneCards, RRID:SCR_002773) and database annotations to identify the following proteins as mitochondrial: HSP90B1, ASS1, MT-CO2 (*Gustafsson et al., 2016*), RPS3 (*Kim et al., 2013*), RACK1 (*Lin et al., 2015*), and ACOT7 (*Bekeova et al., 2019*). This search revealed enrichment of proteins related to energy metabolism and known to localize in mitochondria particularly among upregulated genes. We also queried the Human Lysosome Gene Database (http://lysosome.unipg.it/, *Brozzi et al., 2013*) to identify lysosomal proteins. In ShinyGO, networks of GO biological processes generated using a P-value cutoff for FDR of 0.01, and displaying top 20 most significant terms, with an edge cutoff of 0.2 were also examined to inform the functional categories that most accurately define both down- and upregulated proteins. PPI and GO biological processes networks are shown in *Figure 1—figure supplements 2* and *3*.

## Retrospective assessments of life events and stress

A subset of participants with noteworthy patterns of single-hair greying and reversal were asked to complete a retrospective stress assessment (*Figure 3—figure supplement 1*), completed 1–4 months after hair collection in two individuals (one male, one female). The life event calendar (LEC) methodology increases the reliability and validity of recall in retrospective assessments (*Belli, 1998*). The use of timeline results in higher accuracy and lowers underreporting as compared to traditional questionnaires (*van der Vaart, 2004*). The retrospective psychosocial stress assessment in the present study is an adaptation of LECs, more similar to timelines, which measures one behavioral construct – here 'stress' – during a short reference period (*Glasner and van der Vaart, 2009*; *Sobell et al., 1988*). In our instrument, participants first position landmark events in time (in this case, the most stressful event or period, and the least stressful), and then link other events to these landmark events, a method referred to as sequencing (*Belli, 1998*). Additionally, the visual calendar (see *Figure 3—figure supplement 1*) encourages top-down and parallel retrieval of memories (*Glasner and van der Vaart, 2009*), which also contributes to overall accuracy.

In the retrospective assessment, participants are first asked to identify the most stressful event or period over the last 12 months and to position it in time along the physical timeline, using their electronic calendar and objective dates, and assign it '10' on the graph. This first positioned event acts as a landmark event from which the other events can then be sequentially linked (*Belli, 1998*). Participants then identified the least stressful event or period and assigned it '0' on the physical timeline, acting as another landmark event. Participants then indicated 2–6 additional particularly stressful events or periods, assigned them scores ranging from most stressful to least stressful (10 and 0, respectively), marked them on the timeline, and connected these events with a line that best illustrates their stress levels over the past year. The instrument not only asks the participant to mark their stress levels, but also to briefly name/describe each event, which can help with recall of the exact stressor and its intensity, and also allows participants to match up an event with an exact calendar date, enhancing the timing accuracy of events. Stress graphs were then digitized by aligning the retrospective assessment to a grid printed on transparency film with 0.25 unit resolution (number of possible values = 40 units total, from 0 to 10), and the resulting digital values were plotted (Prism 8). To align stress profiles with HPP, digitized stress profiles were aligned with hairs from the same

participant using dates of collection and assuming a hair growth rate of 1 cm/month (*LeBeau et al., 2011*). Each hair segment can then be mapped to specific weeks or month along the stress profile.

## Hair shaft mtDNA quantification

Dark and white hairs (n = 10 per person per color) were collected from the same two individuals whose hairs were analyzed by electron microscopy (African American male, Caucasian male). The follicle and proximal portion (2 cm segment) of the hair shaft were sectioned and separately lysed in 200 µL of lysis buffer containing 500 mM Tris, 1% Tween 20, 20 µg/µl Proteinase K incubated for 10 hr at 55℃, followed by 10 min at 95℃ as described previously (*Picard et al., 2012*). Hair follicles were fully digested whereas the more robust proteinaceous hair shafts were only partially digested, such that the quantified mtDNA abundance is likely an underestimation of the total DNA amount per unit of hair shaft. In addition, nuclear DNA is rapidly degraded by endonucleases and virtually absent in the hair shaft (*Fischer et al., 2011*). We therefore focus our analysis of genomic material in the hair shaft to mtDNA.

The number of mtDNA copies per nucleated cell (mtDNA copy number, mtDNAcn) was measured by real-time quantitative polymerase chain reaction (qPCR) using a duplex Taqman reaction to amplify both mitochondrial (ND1) and nuclear DNA (B2M, single-copy gene) amplicons. The primer sequences are: (ND1-Fwd: GAGCGATGGTGAGAGCTAAGGT, ND1-Rev: CCCTAAAACCCGCCACA TCT, Probe: HEX-CCATCACCCTCTACATCACCGCCC-3IABkFQ. B2M-Fwd: CCAGCAGAGAA TGGAAAGTCAA, B2M-Rev: TCTCTCTCCATTCTTCAGTAAGTCAACT, Probe:FAMATGTGTCTGGG TTTCATCCATCCGACA-3IABkFQ) obtained from IDTdna.com. qPCR was performed on *QuantStudio 7 Flex Real-Time PCR System* (Life Technologies QuantStudio 7 Real Time PCR System, RRID: SCR_020245) (Applied Biosystems, Foster City, CA). Cycling conditions were as follows; 1 cycle of 50℃ for 2 min, 95℃ for 20 s, followed by 40 cycles of 95℃ for 1 s, 60℃ 20 s.

For plucked hair follicles, all ND1 and B2M Cts were in the dynamic range of the assay and used to compute mtDNAcn from the ΔCt. All measures were performed in triplicates and the average Ct values taken for each sample. The mean C.V. for ND1 was 0.67% in both shafts and follicles, and for B2M 0.52% in follicles. mtDNAcn was calculated as $2^{\Delta Ct}$ (ND1 Ct - B2M Ct), and multiplied by two to account for the diploid nature of the nuclear genome.

## Mathematical modeling of greying dynamics across the lifespan

To simulate hair greying across the lifespan, a linear mixed effect model with random intercept and slopes to account for the stochastic process of hair greying was implemented in R (R Project for Statistical Computing, RRID:SCR_001905) (*R Development Core Team, 2010*). This interactive implementation is available at https://timrain.shinyapps.io/hair (Shiny, RRID:SCR_001626). We first hypothesized a potential mechanism in which individual hairs are affected by a summation of effects from a random aging factor accumulating over time, random stress factor and random initial greying loading, thus creating variation between hairs within an individual. Once the hair has passed a pre-specified threshold, the hair transitions to grey (Figure 5B). This model includes 17 parameters listed in *Supplementary file 4*, each of which can be adjusted to simulate various effects on individual hairs in relation to the aging process, including one or two stress exposure periods with customizable intensity and duration.

Scaling this model to hair populations with thousands of hairs, the simulation reports trajectories of greying for individual hairs, as well as a graph with the population distribution of white hairs (shown as frequency distributions) for a theoretical scalp. First, we simulated the average greying trajectory based on data indicating that the average age of onset for greying is age 35 and that white hairs reach a 40% population frequency at age 65 (*Panhard et al., 2012*). This established a set of default parameters that yielded the greying trajectory shown in Figure 5C. We then simulated two hypothetical scenarios reflecting the total hair population for individuals who accumulate grey hairs at different rates, termed early and late greyers. These variable greying patterns were found to be generated by changing only one parameter, Sigma1, the standard deviation (across HFs) of the rate at which the aging factor increases over time.

Additionally, the model also simulated greying reversal, beginning with the parameters of the average greyer and then including also the stress parameters. To show the effect of stress on hair greying, we simulated two stressful periods starting at age 20 and then again at age 50, with equal

intensity and duration. At age 20 the aging factor increases due to the stress but does not induce grey hair as the aging factor is still below the threshold (Figure 5E). On the other hand, at age 50 the same intensity and duration of the stressor will tend to induce additional greying as the aging factor for some hairs increases past the threshold, and then upon the end of the stressor, the aging factor could decrease past the threshold and thus the hair would undergo reversal (i.e. repigmentation) to its original color (Figure 5F).

An alternative model was considered to explore potential mechanisms for hair transitioning and reversal in response to stress. Specifically, we considered a mechanism in which the *rate* of accumulation in the aging factor increases during a period of stress (as opposed to our final model where stress causes a stepwise increase in aging factor) and then returns to the original rate following the end of the stressor. In this scenario, the threshold remains constant. This mechanism can be rejected because although it adequately simulates hair greying, once a hair has crossed above the threshold, if the stressor only affects the slope, it is not possible for a hair to return below threshold and undergo reversal (*Figure 5—figure supplement 2*).

To simulate the graying process for a hypothetical person based on our hypothesized mechanism, we posited a linear mixed model for the $i$ th ($i = 1, \ldots, n$) hair with two fixed effects ($\beta_1$ aging factor rate and $\beta_2$ stress sensitive rate) and three random effects ($b_{i0}$ for graying loading at age 0, $b_{i1}$ for aging factor rate and $b_{i2}$ for stress sensitive rate). To ensure positivity in age and accumulating stress, the model involves only the absolute value of each random effect.

$$Grey\,in\,Loading_{\{i,age\}} = |b_{i0}| + (|b_{i1}| + \beta_1)age + (|b_{i2}| + \beta_2)Accumulating\,Stress_{\{age\}} + e_{\{i,age\}}$$

where *AccumulatingStress* is defined as:

$$Accumulating\,Stress_{age} = \sum_{a=age-Window\,Width}^{age} stress_a$$

The three random effects follow a multivariate normal:

$$(b_{i0}, b_{i1}, b_{i2}) \sim N(0, G)$$

with covariance structure:

$$G = \begin{bmatrix} \sigma_0^2 & \rho_{01}\sigma_0\sigma_1 & \rho_{02}\sigma_0\sigma_2 \\ \rho_{01}\sigma_1\sigma_0 & \sigma_1^2 & \rho_{12}\sigma_1\sigma_2 \\ \rho_{02}\sigma_2\sigma_0 & \rho_{12}\sigma_2\sigma_1 & \sigma_2^2 \end{bmatrix}$$

All the correlations $\rho_{01}, \rho_{02}, \rho_{12}$ in the simulation are constrained to be positive. When the aging factor of hair $i$ reaches a predefined threshold, the $i$ th hair will turn white. The source code is available at https://github.com/junting-ren/hair_simulation (*Ren, 2021*; copy archived at swh:1:rev: 3a19705969bfca7edc98651c1dd973ca7ae3b23d, RRID:SCR_002630).

## Statistical analyses

An ordinary one-way ANOVA with Tukey's multiple comparison test was used to compare the number of melanin granules per $\mu m^2$, granule size, granule intensity, and relative intensity of the cortex in the dark and white hairs, and to compare the pigmentation intensity across head regions. To compare the rate of change in pigmentation per day between greying and reversal hairs, points of transitions visually estimated were used to derive a slope for each greying or reversal segment, which were compared using an unpaired t test.

A Mann-Whitney test was used to compare mtDNA levels in dark and white hair shafts and mtDNA copy number in dark and white hair follicles.

For univariate and multivariate analyses of proteomic signatures, protein abundance levels were processed in R using the Metaboanalyst 3.0 platform (MetaboAnalyst, RRID:SCR_015539) (*Chong et al., 2019*) as unpaired data. The data was mean-centered and log transformed prior to statistical analyses and missing (low abundance) values were imputed by half of the lowest value for the group (dark, white). Significance was established at an FDR level 0.05 and fold changes calculated using ANOVA. Partial least square discriminant analysis (PLS-DA) was used to extract meaningful features that distinguish dark and white hairs and visualize groups of hairs or segments

along the same hair. Two different strategies were used to generate protein lists subsequently queried for their functional significance: (i) For dark vs white comparisons where the whole model is meaningful, the variable importance in projection (VIP) scores for each protein were extracted and used to select the top 40 most influential proteins (*Figure 1—figure supplement 4C*); (ii) For analyses of segments along the hair with greying followed by reversal, the factor loadings for each protein were extracted separately for components 1 and 2, and the top 20 positive and 20 negative proteins were selected for further analysis. Protein lists derived from both strategies were then used for functional enrichment analysis in ShinyGO and STRING as described above.

For the data displayed in Figure 3, we measured the strength of the temporal relationship between time series on the same scale (e.g. intensity measures of two hairs) by calculating the mean squared differences. The overall strength of temporal relationship among multiple time series is measured by the sum of all pairwise mean squared differences. To measure the strength of the temporal relationship between time series on different scales (specifically, intensity of color for a hair and rated level for stress levels) we calculated Pearson's correlation. To provide a reference distribution for comparison, we conducted 1000 random permutations of the data in each instance. For Figure 3A and B, each permutation involved simulating an equivalent number of hairs that transition (three hairs in 3A; two hairs in 3B). Each simulated hair includes a randomly selected transition placed at a random time point, with resampled noise before and after the transition. Two results are reported in each case. For the first, the library of transitions was taken from the transition segments of each observed hair, regardless of which direction (dark to white or white to dark) that transition was in. The probability of each direction of transition was determined by the overall rate of each transition direction from the observed data. The second reported result is based on a similar analysis, but the library of transitions included only the hairs that underwent the same directional change (dark-to-white in 3A; white-to-dark in 3B). The noise for uses of resampling was taken from hairs from the same subject after subtracting a smooth function, and resampling was done in segments of length-100 increments to maintain proper temporal correlation patterns. For Figure 3D, under the null hypothesis of no relationship between the hair intensity and stress pattern, each permutation involved choosing a random time point, splitting the stress pattern at that point and rejoining it by concatenating the two segments in the alternative order.

## Results

### Mapping HPPs

To overcome the lack of methodology to map pigmentary states and age-related greying transitions, we developed an approach to digitize HPPs at high resolution across the length of single human HSs. Combined with known hair growth rates on the scalp (~1.0–1.3 cm per month *LeBeau et al., 2011*), this approach provides a quantifiable, personalized live bioarchive with the necessary spatio-temporal resolution to map individualized HPPs and greying events along single hairs, and to link HPPs to specific moments in time with unprecedented accuracy. Using this methodology, similar to dendrochronology where tree rings represent elapsed years (*Douglass, 1928*), hair length reflects time, and the HS length is viewed as a physical time scale whose proximal region has been most recently produced by the HF, and where the distal hair tip represents weeks to years in the past, depending on the HS length.

To examine HPPs in human hairs, we plucked, imaged, digitized, and analyzed hairs (n = 397) from 14 healthy donors (*Figure 1A*) (see Materials and methods for details). Three main pigmentation patterns initially emerged from this analysis: (i) Hairs with constant high optical density (*Dark*), (ii) Hairs with constant low optical density (*White*); (iii) Initially dark hairs that undergo a sharp greying transition from dark to white over the course of a single growing anagen phase of the hair follicle growth cycle (*Transition*) (*Figure 1B–C*). Dark-to-white transitions demonstrate the existence of rapid depigmentation events within a single anagen hair cycle (*Paus and Cotsarelis, 1999*; *Slominski et al., 2005*). We confirmed that compared to dark hairs still harboring their 'young' pigmentary state, the HFPU of 'aged' white HFs from either African American or Caucasian individuals are practically devoid of pigment (*Figure 1D*), which is consistent with the finding of previous studies (*Cho et al., 2014*). Whereas dark hairs contain melanin granules dispersed throughout the hair cortex when observed by electron microscopy, white hairs from the same individuals show a near

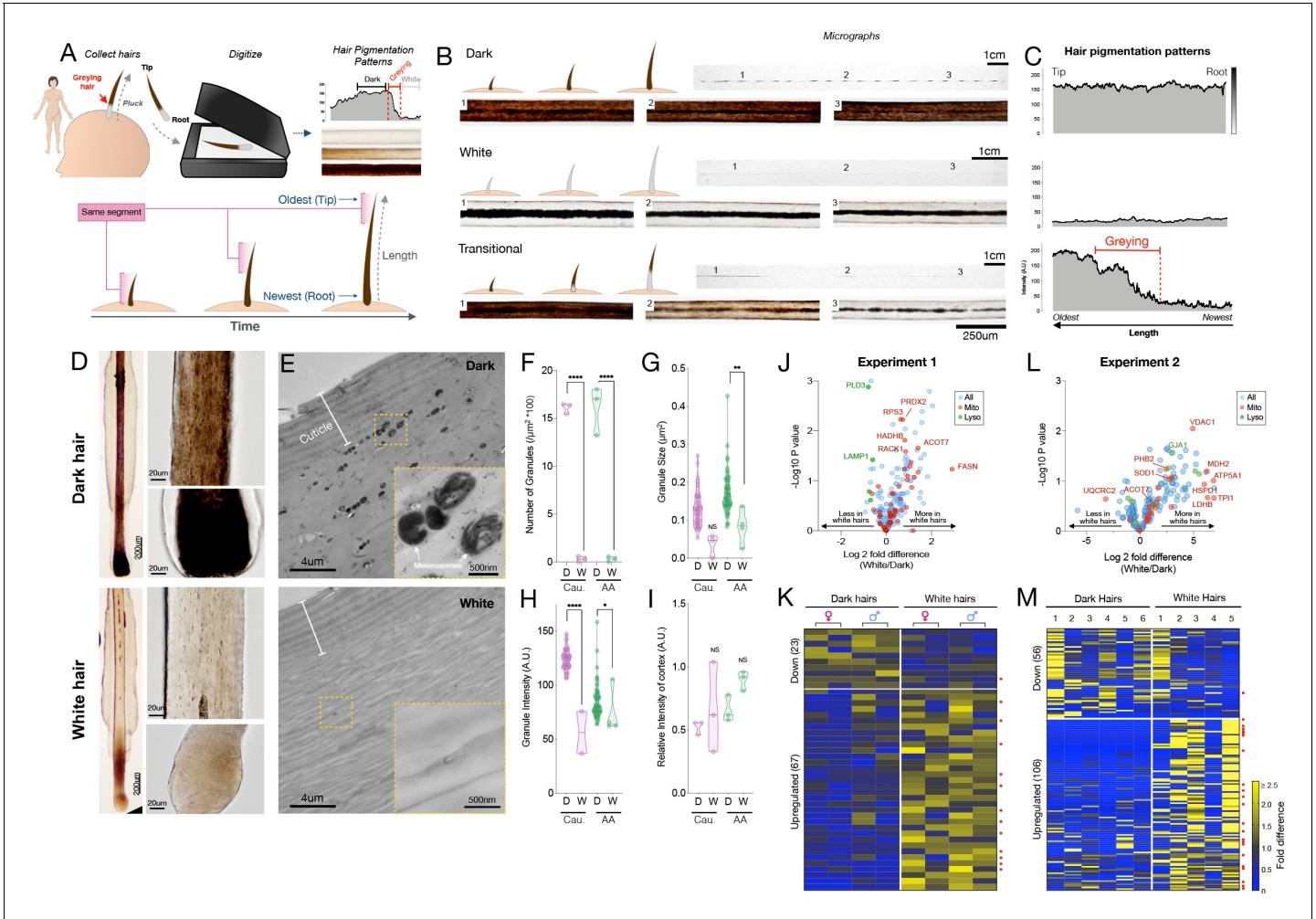

**Figure 1.** Quantitative analysis of human hair pigmentation patterns, greying, and associated proteomic changes. (**A**) Diagram illustrating hair growth over time, method of hair collection, digitization, and hair pigmentation pattern (HPP) methodology. (**B**) Dark, white, and hairs undergoing natural age-related transitions from the younger dark state to the older white state at macroscopic and microscopic resolution. (**C**) Digitized HPPs for the hairs shown in (**B**). (**D**) Bright field microscopy images of hair follicles from plucked dark (top-panel) and white hair (bottom-panel) from the same Caucasian male individual illustrating the loss of pigmentation in the hair follicle pigmentary unit (HFPU). (**E**) Electron microscopic images of dark (*top*) and white (*bottom*) scalp hairs from a Caucasian male showing absent melanin granules in white hairs. (**F**) Quantification from dark (**D**) and white (**W**) hairs (n = 3 segments from each) from a Caucasian (Cau.) male and African-American (AA) male of melanin granule abundance, (**G**) size and (**H**) darkness. (**I**) Overall electron density of the hair matrix (excluding granules) (N.S.). (**J**) Volcano plot comparing dark and white hair proteomes and (**K**) heatmap of down- (<0.8 fold) and up-regulated (>1.5 fold) proteins that were detected in all samples (n = 90) for experiment 1 (duplicates of dark/white hairs from one female, one male, n = 8 samples). (**L**) Volcano plot and (**M**) heatmap for all proteins detected in ≥3 samples (n = 192) from experiment 2 (dark and white hairs from six individuals, n = 6 dark and 5 white hairs). Proteins annotated as mitochondrial (Mitocarta2.0) and lysosomal (The Human Lysosome Gene Database) are highlighted. Red dots to the right of heatmaps indicate mitochondrial proteins. *p<0.05, **p<0.01, ****p<0.0001 from one-way ANOVA, Tukey's multiple comparisons.

The online version of this article includes the following source data and figure supplement(s) for figure 1:

**Source data 1.** Source data for *Figure 1*.
**Figure supplement 1.** Sensitivity analysis of results from LC-MS:MS hair shaft proteomics experiment 2.
**Figure supplement 1—source data 2.** Source data for *Figure 1—figure supplement 1*.
**Figure supplement 2.** Functional enrichment analysis of up- and down-regulated proteins in white hair shafts relative to dark from experiment 1.
**Figure supplement 3.** Functional enrichment analysis of up- and down-regulated proteins in white hair shafts relative to dark in experiment 2.
**Figure supplement 4.** Partial least square discriminant analysis (PLS-DA) of dark and white hairs from experiment 2.
**Figure supplement 4—source data 1.** Source data for *Figure 1—figure supplement 4*.
**Figure supplement 5.** Ultrastructure of melanosomes and dense granules in human hair shafts.

complete (>98%) absence of melanin, with the few retained melanin granules, when present, being smaller, less dense, and at times vacuolated, a potential response to oxidative stress (*Tobin, 2009*; *Figure 1E–I*, see *Figure 1—figure supplement 5* for high-resolution images of mature melanosomes). The digitization of HPPs thus reflects the presence of melanosomes within the HS, and rapid greying events are marked by the loss of melanosomes.

## Proteomic alterations in white hairs

To gain molecular insight into the greying process, we performed a comprehensive proteomic analysis comparing dark and white HS. Recent work suggests that depigmentation is associated with the upregulation of lipid synthesis enzymes in the HS (*Franklin et al., 2020*). Moreover, in depigmented hairs, the abnormal diameter/caliber of the hair fiber, growth rate, presence/absence of HS medulla as well as the (dis)continuity and diameter of the medulla along the hair length (*Van Neste, 2004*) imply multiple potential proteomic alterations associated with depigmentation. In addition, melanogenesis involves high levels of reactive oxygen species, but dark HFs are equipped with multiple antioxidant mechanisms (e.g.[*Tobin, 2009*]). Thus, the proteomic features of HSs may provide interpretable information about molecular changes associated HF greying.

Protein extraction and LC-MS/MS methods optimized from a previous protocol (*Adav et al., 2018*) were used to process the unusually resistant proteinaceous matrix of the hair shaft and to handle the overly abundant keratin proteins over other potential proteins of interest (see Materials and methods for details). Two independent experiments were performed. Experiment 1: matched dark and white hairs collected at the same time from two closely age- and diet-matched individuals (one female and one male, both 35 years old, each dark and white HS measured twice, total n = 8); and Experiment 2 (*validation*): n = 17 hair segments from seven different individuals (four females and three males).

In the first experiment, we were able to extract and quantify 323 proteins (>75% of samples) from single 2-cm-long HS segments. Compared to dark HS collected at the same time from the same individuals, white hairs contained several differentially enriched (upregulated) or depleted (downregulated) proteins (*Figure 1J–K* see *Supplementary file 1* for complete list) on which we performed GO (Gene Ontology) and KEGG (Kyoto Encyclopedia of Genes and Genomes) enrichment analysis and explored their protein-protein interaction networks (*Figure 1—figure supplement 2*). The protein networks for both downregulated (<0.8 fold, n = 23) and upregulated (>1.5 fold, n = 67) proteins contain significantly more interactions than expected by chance (p<0.00001, observed vs expected protein-protein interactions). Thus, coherent groups of functionally related proteins are differentially expressed in white hairs, from which two main patterns emerged.

The first main pattern relates to protein biosynthesis and energy metabolism. A large fraction (34.3%) of upregulated proteins in white hairs was related to ribosome function, protein processing, and associated cytoskeletal proteins. Upregulation of the machinery responsible for protein synthesis and amino acid metabolism included the ribosomal protein RPS15A, which is known to localize to mitochondria. Of all upregulated proteins in white hairs, 26.8% were known mitochondrial proteins, based on MitoCarta2.0 and others sources (*Calvo et al., 2016*). These proteins are involved in various aspects of energy metabolism, including substrate transport (carnitine palmitoyltransferase 1A, CPT1A; malonate dehydrogenase 1, MDH1), respiratory chain function (Complex III subunit 1, UQCRC1), and catecholamine homeostasis (Catechol-O-Methyltransferase, COMT). White hairs also contained more proteins involved in glucose (glucose 6-phosphate dehydrogenase, G6PD; phosphoglycerate kinase 1, PGK1) and lipid metabolism located in either the mitochondria or cytoplasm (fatty acid synthase, FASN; acyl-CoA thioesterase 7, ACOT7; mitochondrial trifunctional enzyme subunit beta, HADHB) or in peroxisomes (acyl-CoA acyltransferase 1, ACAA1). The metabolic remodeling in white hairs is consistent with the established role of mitochondria and metabolic regulation of hair growth and maintenance in animal models (*Flores et al., 2017*; *Kloepper et al., 2015*; *Singh et al., 2018*; *Vidali et al., 2014*), and possibly consistent with hair anomalies reported in human patients with mitochondrial disease (*Silengo et al., 2003*). The upregulation of energy metabolism may subserve the likely increased energy demands in depigmented hairs. However, our data and those of others (*Franklin et al., 2020*) implicate the upregulation of specific mitochondrial proteins involved, not necessarily in global energy metabolism, but in specific metabolic activities such as amino acid and lipid biosynthesis.

A second less robust pattern relates more directly to melanosome biology. In line with the lysosomal origin of melanosomes that are largely absent in depigmented HS (*Slominski et al., 2005*), several lysosomal proteins (PLD3, CTSD, HEXB, and LAMP1) were downregulated in white hairs, consistent with previous literature (*Franklin et al., 2020*). White hair shafts also showed a depletion of six main keratins (see *Figure 1—figure supplement 2*), likely because greying can affect the nature of keratinocytes proliferation (*Van Neste and Tobin, 2004*), of proteins associated with exocytosis, such as ITIH4 and APOH (potentially involved in the secretion of melanosomes from melanocytes to keratinocytes), as well as proteins associated with mitochondrial calcium transmembrane transport. Interestingly, calcium exchange between mitochondria and the melanosome is required for melanin pigment production in melanocytes (*Zhang et al., 2019*), and calcium signaling promotes melanin transfer between melanosomes and keratinocytes (*Singh et al., 2017*).

Finally, canities-affected white HFs also showed upregulation of antioxidant proteins, specifically those localized to mitochondria and cytoplasm (superoxide dismutase 1, SOD1; peroxiredoxin 2, PRDX2), in line with the role of oxidative stress in HS depigmentation (*Arck et al., 2006*; *Wood et al., 2009*). Alterations among these individual metabolic and mitochondrial enzymes were further reflected in KEGG pathways functional enrichment analyses indicating a significant enrichment of metabolic pathways including carbon metabolism and fatty acid synthesis, amino acids, and oxidative phosphorylation (see below).

## Validation of greying-associated proteomic signatures

To independently validate these results, we extended this analysis to white and dark HS from six individuals (three males, three females, range: 24–39 years) analyzed on a separate proteomic platform and in a different laboratory. In this experiment, a total of 192 proteins ($\geq$3 samples) were quantified from 1cm-long HS segments. This dataset showed a similar trend as the first analysis toward a preferential overexpression of proteins with greying (55% upregulated vs 29% downregulated in white HS) (*Figure 1L–M*, see *Supplementary file 2* for a complete list). The most highly upregulated proteins included mitochondrial components such as the voltage-dependent anion channel 1 (VDAC1), a subunit of ATP synthase (ATP5A1), and a regulator of mitochondrial respiratory chain assembly (Prohibitin-2, PHB2). Again, the antioxidant enzyme SOD1 was enriched in white relative to dark HSs.

To examine the possibility that these relative upregulations are driven by a global downregulation of highly abundant housekeeping proteins, we analyzed the intensity-based absolute quantification (iBAQ) data for each sample. This confirmed that the housekeeping proteins, including keratins and keratin-associated proteins, were not downregulated in white hairs, but generally unchanged or slightly upregulated. Moreover, as a whole, upregulated proteins formed a coherent protein-protein interactions cluster (p<0.00001) and pathway analysis again showed a significant enrichment of carbon metabolism, glycolysis/glucogenesis, pyruvate metabolism, and amino acid synthesis pathways in white relative to dark HS (*Figure 1—figure supplement 3*, Figure 4E). As in experiment 1, these upregulated pathways all indicate metabolic remodeling in white hair follicles. On the other hand, proteins downregulated in white HSs were related to cholesterol metabolism, complement-coagulation cascades, and secretory processes shared with immune/inflammatory pathways (Figure 4E). The downregulation of secretory pathways is again consistent with reduced transfer of pigmented melanosomes from the melanocytes to the keratinocytes.

To verify the robustness of these findings using an alternative analytical approach, we built a simple partial least square discriminant analysis (PLS-DA) multivariate model, which provided adequate separation of white vs dark HS (*Figure 1—figure supplement 4*). Simple interrogation of this model to extract the features (i.e. proteins) that maximally contribute to group separation yielded a set of proteins enriched for estrogen signaling pathways, complement and coagulation cascades, as well as metabolic pathways including NAD$^+$/NADH, cholesterol, pyruvate, and carbon metabolism, similar to results above. Interestingly, we also identified 13 proteins that were undetectable in any of the dark HS (either not expressed, or below the detection limit), but consistently present in white HS (*Supplementary file 3*). These proteins are either newly induced or experience a substantial upregulation with greying (fold change tending toward infinity). A separate functional enrichment analysis for these induced proteins also showed significant enrichment for the same aging-related metabolic pathways as for the upregulated protein list: glycolysis/glucogenesis, carbon, pyruvate, and cysteine and methionine metabolism (all p<0.001).

These converging proteomics data, which are consistent with previous findings (*Franklin et al., 2020*), support a multifactorial process directly implicating metabolic changes in human hair greying (*Paus, 2011*). Moreover, given that metabolic pathways are rapidly and extensively remodeled by environmental and neuroendocrine factors – that is, they naturally exhibit plasticity – these data build upon previous proteomic evidence to show that human hair greying could be, at least temporarily, reversible.

## Human hair greying is, at least temporarily, reversible

Our analysis of HPPs in healthy unmedicated individuals revealed several occasions whereby white hairs naturally reverted to their former dark pigmented state. This phenomenon was previously reported only in a handful of case reports, with only a single two-colored HS in each case (O'Sullivan, 2020). Here, we document the reversal of greying along the same HS in both female and male individuals, ranging from a prepubescent child to adults (age range 9–39 years), and across individuals of different ethnic backgrounds (1 Hispanic, 8 Caucasian, 1 Asian). This phenomenon was observed across frontal, temporal, and parietal regions of the scalp (*Figure 2A*), as well as across other corporeal regions, including pubic (*Figure 2B*) and beard hairs (*Figure 2C*). The existence of white HS undergoing repigmentation across ages, sexes, ethnicity, and corporeal regions documents the reversibility of hair greying as a general phenomenon not limited to scalp hairs. Nevertheless, we note that this phenomenon is limited to rare, isolated hair follicles. As their occurrence will probably go unnoticed in most cases, it is difficult to assess the true incidence of these repigmentation events. Over an active recruitment period of 2.5 years, we were only able to identify 14 participants, indicative of the rarity of this phenomenon.

Moreover, more complex HPPs with double transitions and reversions in the same HS were observed in both directions: HS undergoing greying followed by rapid reversal (*Figure 2D*), and repigmentation rapidly followed by greying (*Figure 2E*). Importantly, both patterns must have occurred over the course of a single anagen (growth) phase in the hair growth cycle, implicating cellular mechanisms within the HFPU. Greatly extending previous case studies of these rare hair repigmentation events, the current study provides the first quantitative account of the natural and transitory reversibility of hair greying in humans.

We understand the emergence of a reverted HS – that is, a HS with a white distal segment but with a dark proximal end – as necessarily having undergone repigmentation to its original pigmented state following a period of time as a depigmented 'old' white hair (*Figure 2F*). In double transition HS with three segments, repigmentation must take place within weeks to months after greying has occurred, producing three distinct segments present on the same hair strand (*Figure 2G*). Microscopic imaging along the length of a single HS undergoing a double transition (greying followed by rapid reversal) can be visualized in *Video 1*, illustrating the dynamic loss and return of pigmented melanosomes within the same HS. A proposed mechanism for such repigmentation events involve the activation and differentiation of a subpopulation of immature melanocytes located in a reservoir outside of the hair follicle bulb in the upper outer root sheath (*Van Neste and Tobin, 2004*), or more likely from transient amplifying melanoblast cells that migrate along the outer root sheath to the interfollicular epidermis (*Birlea et al., 2017*).

Our hair digitization approach also provides the first estimates of the rates of change in pigmentation for HS covering a broad range of initial colors and darkness (*Figure 2H*). Across

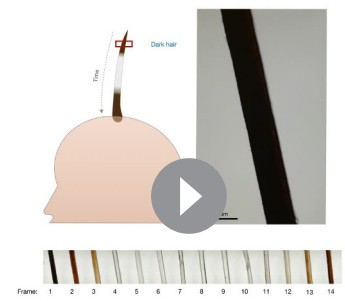

**Video 1.** Microscopic visualization of hair greying and reversal in a single human hair shaft. A greying transition followed by complete reversal in a single hair shaft, imaged from bulb to tip. Pigmentation intensity was dynamically captured on a motorized stage microscope at ×10 magnification from the plucked hair of a 30-year-old Asian participant (hair analyzed in *Figure 2D*).
https://elifesciences.org/articles/67437#video1

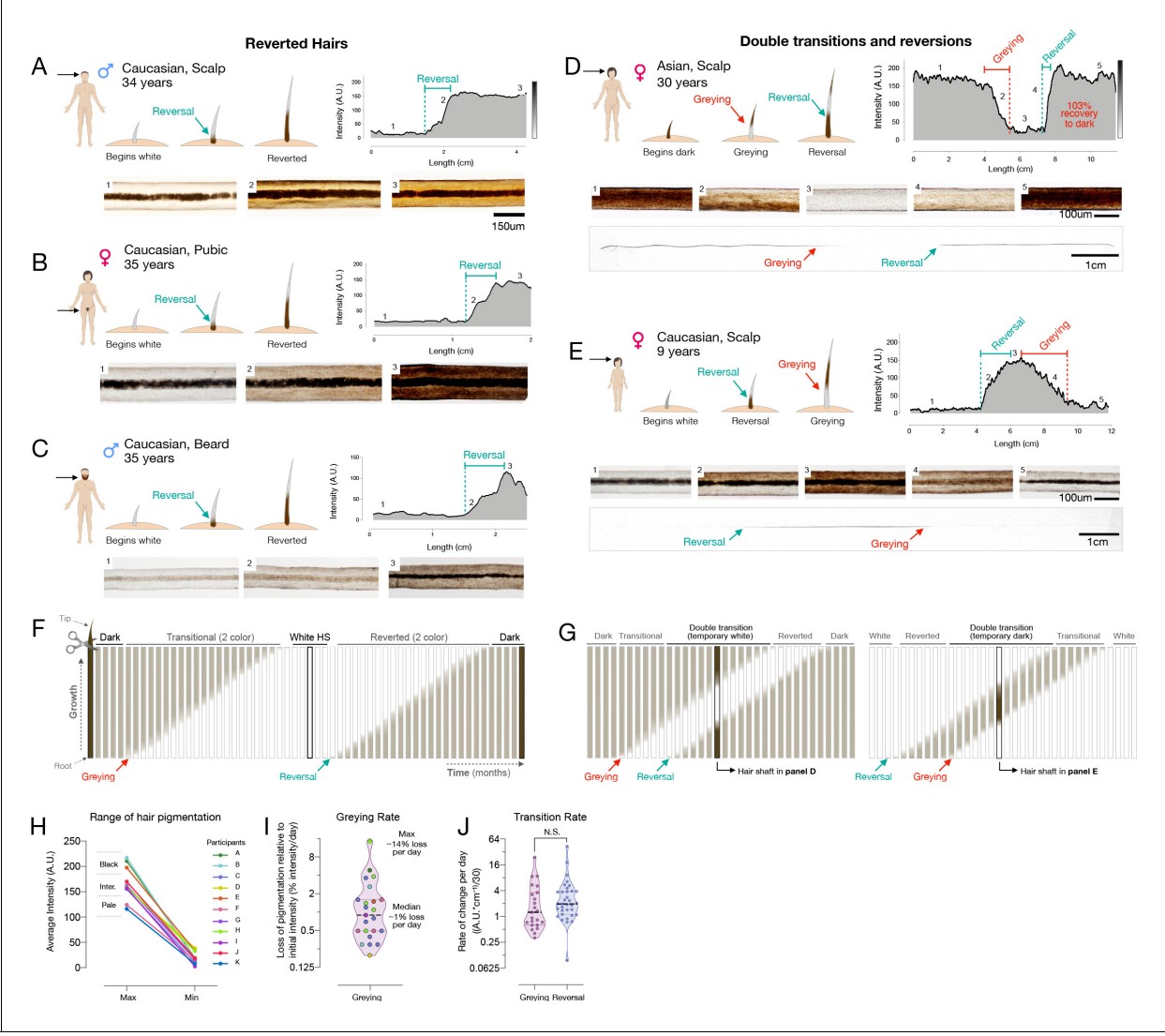

**Figure 2.** Reversal of hair greying across ages and body regions. (**A–G**) Examples of HS greying and reversal including schematic of hair growth (*top left*), digitized HPP (*top right*), and light microscopy images (*bottom*) corresponding to numbered HS segments on the HPP plot. (**A**) Examples illustrating the reversal of greying along the length of scalp, (**B**) pubic, (**C**), and beard human HSs. (**D**) Example of segmental HS with double transitions, including temporary greying and (**E**) temporary reversal from an adult and a child, respectively. See *Figure 2—figure supplement 1* for additional examples and *Video 1* for animation. (**F**) Time course diagram illustrating the progression of a single dark HS undergoing greying followed by reversal back to its original color, and (**G**) closely occurring events of greying and reversal occurring, producing HS with double transitions. (**H**) Average maximum and minimum pigmentation intensity among transitioning hairs from participants with two-colored hairs (n = 11). Hairs with an average maximum intensity >180 A.U. are categorized as high intensity (black), 140–180 A.U. as intermediate intensity, and 100–140 A.U. as low intensity (pale color), indicating that these findings generalize across a range of pigmentation densities. (**I**) Rate of depigmentation per day in greying HS (n = 23), measured from the slope on HPP graphs expressed as % of starting intensity loss per day (assuming growth rate of 1 cm/month). (**J**) Comparison of the absolute rate of pigmentation change per day in greying (n = 23) and reverted (n = 34) HS. (I) and (J) are reported on a log$_2$ scale to facilitate visualization.

The online version of this article includes the following source data and figure supplement(s) for figure 2:

**Source data 1.** Source data for *Figure 2*.

**Figure supplement 1.** Hair pigmentation patterns across a range of individuals.

**Figure supplement 1—source data 1.** Source data for *Figure 2—figure supplement 1*.

individuals, assuming a scalp hair growth rate of 1 cm/month (*LeBeau et al., 2011*), the rates of depigmentation in greying hairs ranged widely from 0.3 to 23.5 units of hair optical density (scale of 0–255 units) per day, corresponding to between 0.2% and 14.4% loss of hair color per day

(*Figure 2I*). The rate at which HS regain pigmentation during reversal was 0.1–42.5 units per day, which is similar (~30% faster on average) to the rate of greying (Cohen's d = 0.15, p=0.59) (*Figure 2J*). Given these rates, the fastest measured transitioning hairs grey and undergo full reversal in ~3–7 days (median: ~3 months). Thus, rather than drifting back toward the original color, repigmentation of white human HS occurs within the same time frame and at least as rapidly as the process of greying itself.

The spectrum of greying transitions and reversals patterns observed in our cohort, including measured rates of repigmentation along individual hairs, is shown in *Figure 2—figure supplement 1*. The HPP results establish the wide range of naturally occurring rates of pigmentary changes in single hairs, which vary by up to an order of magnitude from hair to hair. These data also suggest that reversal/repigmentation may reflect the action of as yet unknown local or systemic factors acting on the HFPU within a time frame of days to weeks.

## Correlated behavior of multiple scalp hairs

We then asked whether the reversal of greying is governed by a process that is unique to each human scalp HF or if it is likely coordinated by systemic factors that simultaneously affect multiple HFs. Participants' scalps were visually inspected to identify two-colored hairs, including both greying transitions and reversal. In our combined cohort, three individuals had multiple two-colored hairs collected at either one or two collection times within a one-month interval. In each case, the multiple two-colored hairs originated from independent HFs separated by at least several centimeters (e.g. left vs right temporal, or frontal vs temporal). If the hairs are independent from each other, the null hypothesis is that different HSs will exhibit either greying or reversal changes at different positions along each hair, and will have independent HPPs. If multiple HSs were coordinated by some systemic factor, then we expect HPPs to exhibit similarities.

In a first 35-year-old female participant with dark brown hair, three two-colored hairs were identified at a single instance of collection. Notably, all three hairs exhibited dark-to-white greying. Moreover, when the HPPs of the three hairs were quantified and plotted, the HPPs followed strikingly similar greying trajectories (r = 0.76–0.82) marked by a similar time of onset of greying, similar HPP intermittent fluctuations (note the rise ~10 cm), and a similar time point where all hairs become fully depigmented (~15 cm) (*Figure 3A*). A permutation test on the similarity of the color trajectories yielded p=0.032, suggesting possible synchrony between different HSs. If our simulation considers only hairs that transition in one direction (from dark to white) this gives p=0.086 (see Materials and methods for details).

In a 37-year-old female participant with brown hair, two transition hairs were identified. The HPPs for both hairs revealed strikingly similar trajectories (r = 0.80), in this case undergoing spontaneous reversal in a near-synchronous manner upon alignment (p<0.001 when considering hairs transitioning in either direction, and similarly, p<0.001 considering only hairs transitioning from white to dark, *Figure 3B*). Thus, these findings extend previous reports in single isolated hairs by providing quantitative accounts of coordinated HS (re)pigmentation across multiple hairs.

Candidate humoral hair pigmentation modulators that could create synchrony in greying or repigmenting hairs include neuropeptides, redox balance, and steroid or catecholamine hormones (*Hardman et al., 2015*; *Paus et al., 2014*; *Tobin and Kauser, 2005*; *Zhang et al., 2020*) that can directly regulate the human HFPU (*Paus, 2011*), impact intrafollicular clock activity (*Hardman et al., 2015*), or regulate the expression of other melanotropic neurohormones in the human HFPU such as α-MSH, ß-endorphin, and TRH (*Gáspár et al., 2011*). These factors must act in parallel with genetic factors that influence inter-individual differences in aging trajectories.

## Hair greying and reversal are linked to psychosocial stress levels

In light of these results, we next applied our HPP method to examine the possibility that psychological stress is associated with hair greying/reversal in humans. Anecdotal case reports and a recent pilot study suggest that psychological stress and other behavioral factors accelerate the hair greying process (*Nahm et al., 2013*; *Peters et al., 2017*), a notion supported by studies in mice demonstrating that adrenergic stimulation by norepinephrine signaling leads to melanocyte stem cell depletion in mice (*Zhang et al., 2020*). However, contrary to mice where this process appears to be irreversible at the single hair follicle level, our data demonstrates that human hair greying is, at least under

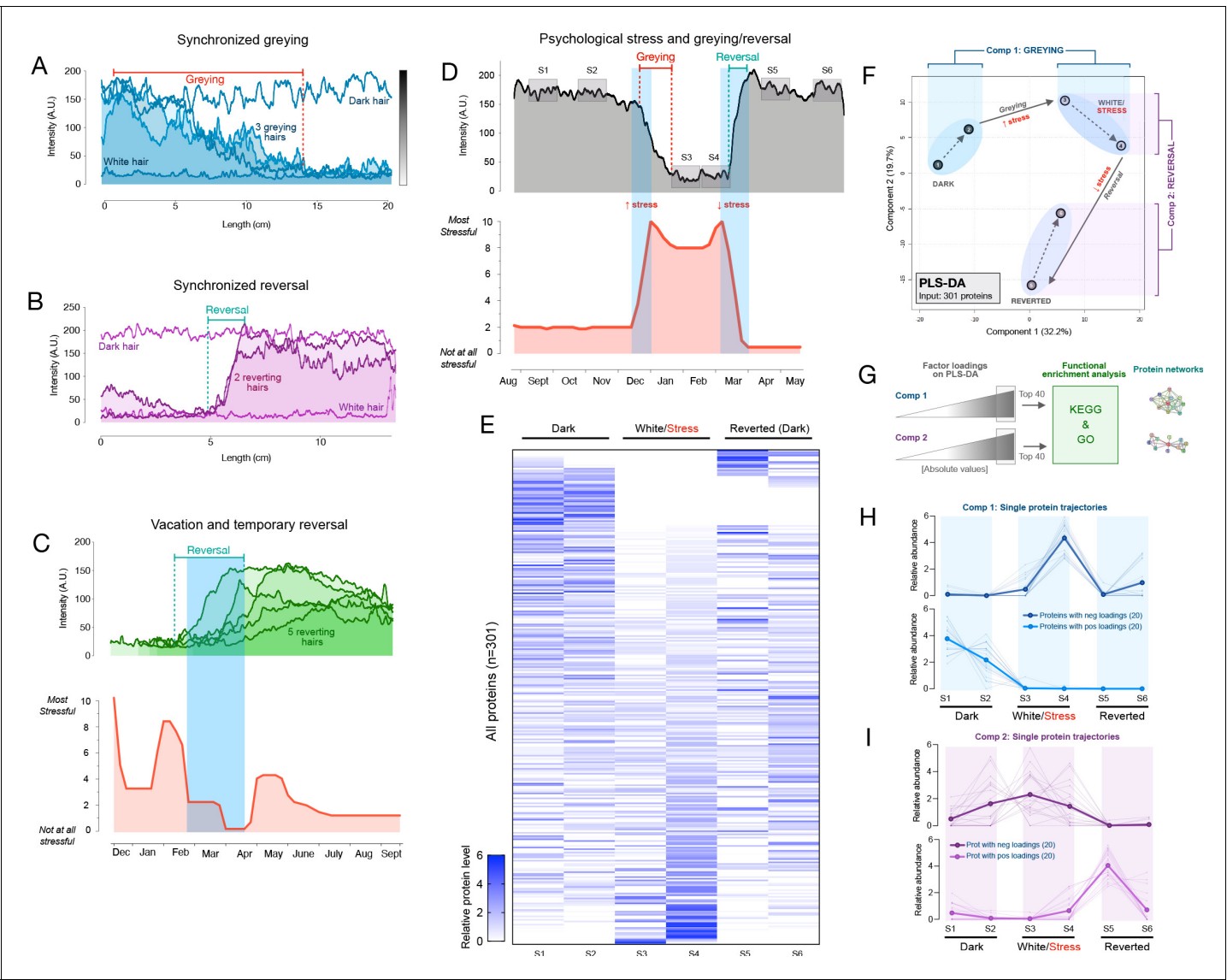

**Figure 3.** Synchronous greying and reversal behavior across multiple hairs and associations with psychosocial stress. (A) In a 35-year-old Caucasian female, multiple HS (n = 3) undergoing greying simultaneously. (B) In a 37-year-old Caucasian female, two bi-color HS collected 2 months apart aligned based on a growth rate of 1 cm/month undergoing reversal nearly simultaneously. In A and B, simultaneously plucked dark and white hairs are plotted for reference. (C) In a 35-year-old Caucasian male, multiple bi-color HS (n = 5) undergoing reversal (top-panel) plotted against time-matched self-reported psychosocial stress levels (bottom-panel). (D) HS from a 30-year-old Asian female with 2 months of self-reported profound perceived stress associated with temporary hair greying and reversal. Note the synchronicity between the increase in stress and rapid depigmentation (i.e. greying), followed by complete (103%) recovery of HS pigmentation (i.e. reversal of greying) upon alleviation of life stress. (E) Heatmap of protein abundance (n = 301) across six segments: two dark prior to stress/greying, two white following greying, two dark segments after reversal. (F) Multivariate PLS-DA model of the six segments from the HS in E, highlighting the model's first and second principal components related to greying and reversal, respectively. Numbers 1 to 6 correspond to HS segments on D. (G) Factor loadings for Components 1 and 2 were used to extract the most significant proteins, subsequently analyzed for functional enrichment categories in KEGG and GO databases, and STRING protein networks. (H) Trajectories of protein abundance from the top Component one and (I) Component two features across the six segments; proteins with positive (*top*) and negative loadings (*bottom*) are shown separately.

The online version of this article includes the following source data and figure supplement(s) for figure 3:

**Source data 1.** Source data for *Figure 3*.

**Figure supplement 1.** Retrospective stress assessment instrument.

**Figure supplement 2.** Single-hair multi-segment analysis.

**Figure supplement 2—source data 1.** Source data for *Figure 3—figure supplement 2*.

some circumstances, reversible. This dichotomy highlights a potential fundamental difference between rodent and human HF biology, calling for a quantitative examination of this process in humans.

As evidence that environmental or behavioral factors influence human hair greying, epidemiological data suggests that smoking and greater perceived life stress, among other factors, are associated with premature greying (*Akin Belli et al., 2016*). Chronic psychosocial stress also precipitates telomere shortening, DNA methylation-based epigenetic age, as well as other biological age indicators in humans (*Epel et al., 2004*; *Han et al., 2019*), demonstrating that psychological factors can measurably influence human aging biology. In relation to mitochondrial recalibrations, psychosocial factors and induced stress can also influence mitochondrial energetics within days in humans (*Picard et al., 2018*) and animals (*Picard and McEwen, 2018*). To generate proof-of-concept evidence and test the hypothesis that psychosocial or behavioral factors may influence greying at the single-HF level, we leveraged the fact that HPPs reflect longitudinal records of growth over time – similar to tree rings – which can be aligned with assessments of life stress exposures over the past year. By converting units of hair length into time, perceived stress levels can be quantitatively mapped onto HPPs in both greying and transitional hairs.

A systematic survey of two-colored hairs on the scalp of a 35-year-old Caucasian male with auburn hair color over a 2-day period yielded five two-colored HS from the frontal and temporal scalp regions. Again, two-colored hairs could either exhibit depigmentation or reversal. Unexpectedly, all HS exhibited reversal. HPP analysis further showed that all HS underwent reversal of greying around the same time period. We therefore hypothesized that the onset of the reversal would coincide with a decrease in perceived life stress. A retrospective assessment of psychosocial stress levels using a time-anchored visual analog scale (participants rate and link specific life events with start and end dates, see Materials and methods and *Figure 3—figure supplement 1* for details) was then compared to the HPPs. The reversal of greying for all hairs coincided closely with the decline in stress and a 1-month period of lowest stress over the past year (0 on a scale of 0–10) following a 2-week vacation (*Figure 3C*).

We were also able to examine a two-colored hair characterized by an unusual pattern of complete HS greying followed by rapid and complete reversal (same as in *Figure 2B*) plucked from the scalp of a 30-year-old Asian female participant with black hair. HPP analysis of this HS showed a white segment representing approximately 2 cm. We therefore hypothesized that this reversible greying event would coincide with a temporary increase in life stress over the corresponding period. Strikingly, the quantitative life stress assessment over the last year revealed a specific 2-month period associated with an objective life stressor (marital conflict and separation, concluded with relocation) where the participant rated her perceived stress as the highest (9–10 out of 10) over the past year. The increase in stress corresponded in time with the complete but reversible hair greying (*Figure 3D*). This association was highly significant (p=0.007) based on our permutation test. Given the low statistical probability that these events are related by chance, life stress is the likely preceding cause of these HS greying and reversal dynamics. These data demonstrate how the HPP-stress mapping approach allows to examine the coordinated behavior of greying and reversal dynamics with psychosocial factors, raising the possibility that systemic biobehavioral factors may influence multiple HFs simultaneously and regulate HPPs among sensitive hairs.

## Single-hair longitudinal proteomic signature

To assess whether rapid greying and reversal events among a single hair are molecularly similar or distinct to those described in the two proteomics experiments above, we dissected six segments (two dark, two white, two reverted) of the single HS in *Figure 3D* and quantified their proteomes as part of Experiment 2. This produced a longitudinal, single-hair, proteomic signature (*Figure 3E*) containing 301 proteins quantified in ≥2 of the six segments. To examine how the proteome as a whole is altered through the greying and reversal transitions associated with psychosocial stress levels, we generated a PLS-DA model with all six segments. Both dark segments clustered together, with similar values on both first and second principal components. The white and reverted segments clustered in separate topological spaces (*Figure 3F*). Greying was associated with a positive shift largely along the first component (Component 1), whereas reversal was associated with a negative shift on the second component (Component 2) and a more modest negative shift in Component 1. We therefore extracted loading weights of each protein on Components 1 and 2 (reflecting each

protein's contribution to group separation) and used the top proteins (n = 20 highest negative, and 20 most positive loadings, total n = 40 per component) to interrogate KEGG and GO databases.

The model's Component 1 (associated with greying) contained proteins that were either (i) not expressed in the dark HS but induced selectively in the white HS segment or (ii) highly abundant in dark segments but strongly downregulated in white and reverted segments (*Figure 3H*, *top* and *bottom*, respectively). In gene set enrichment analysis of Component 1 (greying), the top three functional categories were carbon metabolism, glycolysis/gluconeogenesis, and Kreb's cycle (*Figure 4E*). Component 2 (reversal)-associated proteins exhibited distinct trajectories either peaking in the first white segment or upon reversal (*Figure 3I*) and mapped to pathways related to the complement activation cascade, infectious processes, and Parkinson's and Huntington's disease (*Figure 4E*). In contrast, a null set of hair proteins not contributing to either components exhibited enrichment for extracellular exosomes and cell-cell adhesion that reflect hair shaft biology (*Figure 3—figure supplement 2*), illustrating the specificity of our findings related to greying and reversal. These data indicate that the reversal of greying at the single-hair level is not associated with a complete reversal in the molecular composition of the HS. Rather, some of the proteomic changes in hair greying are enduring despite successful repigmentation.

## Conserved proteomic signatures of hair greying

To systematically examine the overlap among the different proteomic datasets and to derive functional insight into the hair greying process in humans, we then integrated results from the three datasets described above. White HS show consistently more upregulated than downregulated proteins across datasets (2.91-fold in Experiment 1, 1.89-fold in Experiment 2) (*Figure 4A*). This preferential upregulation suggests that the depigmentation process likely involves active metabolic remodeling rather than a passive loss of some pigmentation-related factor. The overlap in the specific proteins identified across dark-white comparisons and among the 6-segments hair is illustrated in *Figure 4B*.

Five proteins were consistently upregulated between experiments 1 and 2. These include three well-defined resident mitochondrial proteins involved in lipid metabolism: CPT1A, which imports fatty acids into mitochondria *Schlaepfer and Joshi, 2020*; ACOT7, which hydrolyzes long-chain fatty acyl-CoA esters in the mitochondrial matrix and cytoplasm (*Bekeova et al., 2019*); and SOD1, which dismutates superoxide anion into hydrogen peroxide ($H_2O_2$) in the mitochondrial intermembrane space (*Okado-Matsumoto and Fridovich, 2001*). The other two proteins include the actin-depolymerizing protein cofilin-1 (CFL1) and the core glycolysis enzyme phosphoglycerate kinase 1 (PGK1) (*Figure 4C*). Interestingly, beyond its role in cytoskeleton dynamics, CFL1 promotes mitochondrial apoptotic signaling via cytochrome c release (*Hoffmann et al., 2019*) and regulates mitochondrial morphology via its effect on actin polymerization-dependent mitochondrial fission (*Rehklau et al., 2017*). And although PGK1 is a cytoplasmic kinase, it was recently demonstrated to translocate inside mitochondria where it phosphorylates and inhibits pyruvate dehydrogenase and Krebs cycle activity (*Nie et al., 2020*). Thus, all five proteins validated across both experiments are linked to mitochondrial energy metabolism, implicating mitochondrial remodeling as a feature of hair greying. Interestingly, all five proteins have also been linked to the biology of melanocytes (*Bracalente et al., 2018*; *Morvan et al., 2012*; *Oh et al., 2014*; *Sumantran et al., 2015*; *Sung et al., 2016*), the source of pigment in the HFPU. The downregulated proteins were keratins, with small effect sizes, and not particularly robust. Analysis of the intensity based on absolute quantification (iBAQ) data confirmed the upregulation of these five mitochondrial proteins, and the absence of substantial changes in the keratins. Together, these data suggest that HS proteome profiling may provide a retrospective access to some aspect of melanocyte metabolism, which opens new possibilities to study HF aging biology.

Since the observed proteomic signatures are related to specific metabolic pathways rather than the typical high-abundance mitochondrial housekeeping proteins, we reasoned that the upregulation of these mitochondrial components unlikely reflects a bulk increase in total mitochondrial content. To investigate this point using an independent method, we quantified mitochondrial DNA (mtDNA) abundance in human HSs by real-time qPCR. Both white and dark HSs contain similarly high levels of mtDNA (*Figure 4D*). The same was true in the follicles of the same hairs (*Figure 4—figure supplement 1*). The similar mtDNA levels between dark and white HSs and HFs increases the likelihood

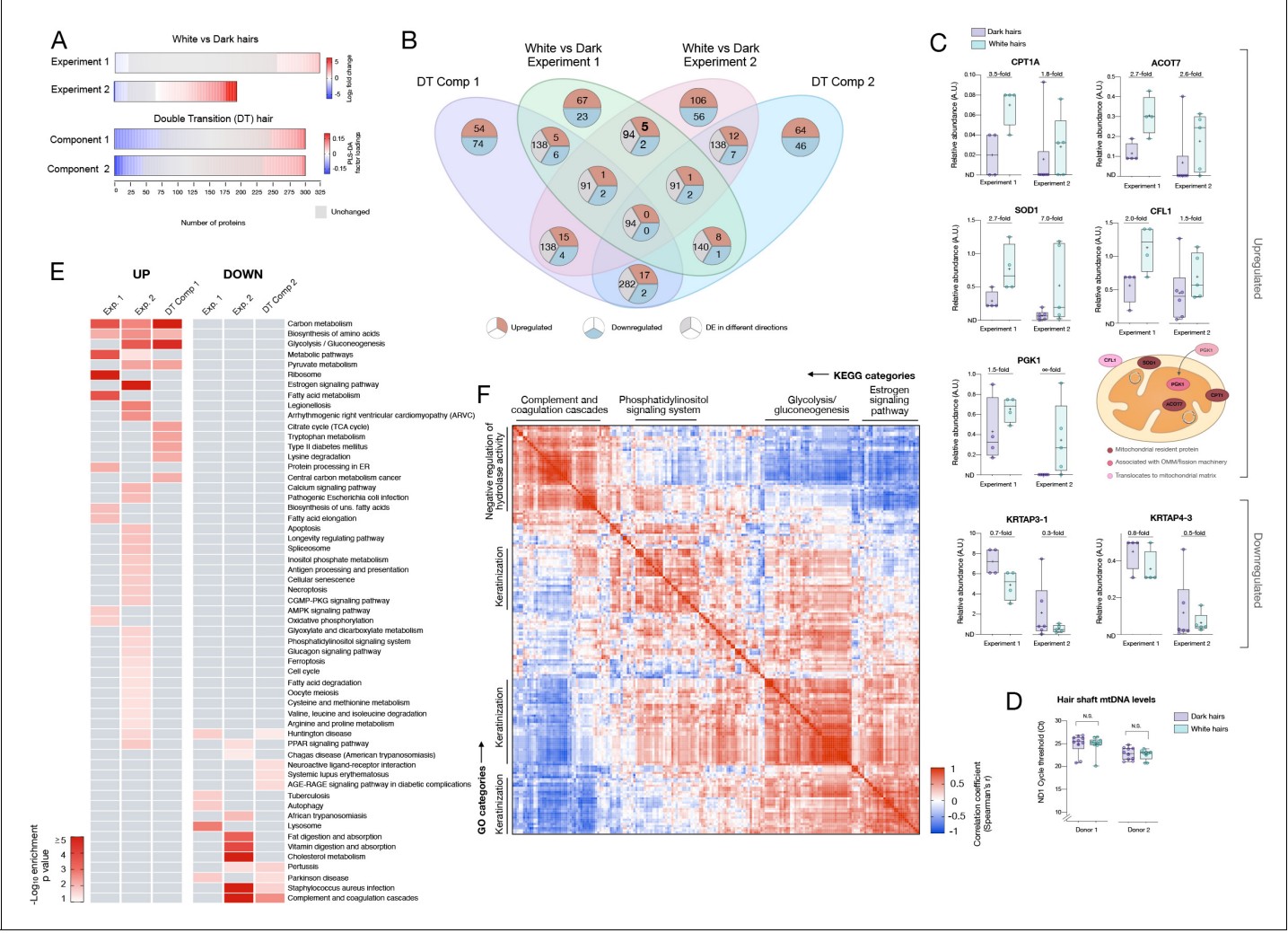

**Figure 4.** Meta-analysis of human hair proteomic findings comparing dark and white hairs. (**A**) Number of downregulated (<0.8 fold, blue) and upregulated (<1.5 fold) proteins across datasets, and unchanged proteins shown in grey. (**B**) Venn diagram illustrating the intersection of datasets. The number of overlapping proteins across datasets that are either consistently down- or upregulated, or proteins not regulated in the same direction, are shown for each area of overlap. (**C**) Individual protein abundance for consistently upregulated (n = 5) and downregulated proteins (n = 2) across experiments 1 and 2 shown are shown as box and whiskers plots, with bars extending from the 25th-75th percentiles, and whiskers from min to max values. Lines indicate the median and '+' signs indicate the mean. Fold difference values are the mean fold differences relative to dark hairs. (**D**) Mitochondrial DNA abundance in human HS of the same two donors as in *Figure 1F–I* (AA male, Cau male). (**E**) Summary of significantly enriched KEGG categories across datasets, for upregulated (*left*) and downregulated (*right*) proteins. (**F**) Correlation matrix (Spearman's r) of all detected proteins (n = 192) in experiment 2, illustrating general human hair protein co-expression across dark and white pigmented states (dark, white). Four main clusters are highlighted and labeled by their top KEGG category. N.S. from Mann Whitney Test.

The online version of this article includes the following source data and figure supplement(s) for figure 4:

**Source data 1.** Source data for *Figure 4*.

**Figure supplement 1.** Mitochondrial DNA copy number in dark and white hair follicles.

**Figure supplement 1—source data 1.** Source data for *Figure 4—figure supplement 1*.

**Figure supplement 2.** Correlation matrix reflection co-regulation of the human hair proteome (n = 192).

that the reported proteomic changes reflect the induction of specific metabolic pathways associated with hair greying rather than bulk increase in mitochondrial mass.

To identify a general proteomic signature of greying hair, we compiled the enrichment scores for KEGG pathways across all datasets (*Figure 4E*). Consistent with the function of the individual proteins identified in both group comparisons (Experiments 1 and 2) and the multi-segment double-transition hair, white HS showed consistent upregulation of carbon metabolism and amino acid

biosynthesis, glycolysis/gluconeogenesis, and general metabolic pathways relevant to aging biology (*Wiley and Campisi, 2016*). Comparatively fewer pathways were consistently represented for down-regulated proteins across independent experiments.

In relation to hair biology in general, our data adds to previous efforts (*Franklin et al., 2020*) and provides a quantitative map of the co-expression among keratin and non-keratin HS proteins across dark and white hairs (*Figure 4F*). Computing the cross-correlations for each protein pair revealed four main clusters among the HS proteome (*Figure 4—figure supplement 2*). As expected for hair, keratins were well-represented and constituted the main GO biological processes category for three of the four clusters. The top KEGG categories included glycolysis and estrogen signaling pathways, which also showed strong co-expression with each other, highlighting potential interaction among endocrino-metabolic processes in relation to human hair pigmentation. In general, the identification of several non-keratin metabolism-related proteins in the HS opens new opportunities to investigate greying pathobiology and to non-invasively access past molecular and metabolic changes that have occurred in the aging HFPU of the dynamically growing hair.

## In silico modeling of hair greying and its temporary reversal

Finally, to narrow the range of plausible mechanisms for the observed age-related greying and reversal events, we developed a simulation model of HPPs. Greying dynamics of an individual's hair population (~100,000 hairs) across the average 80 year lifespan cannot practically be measured. In the absence of such data, we propose here a mathematical model to simulate hair greying trajectories across the human lifespan (*Figure 5A*, available online, see Materials and methods for details) as has been attempted previously for hair growth cycles (*Halloy et al., 2000*). As basic tenets for this model, it is established that (i) the onset of human hair greying is not yet underway and rarely begins in childhood, (ii) greying routinely starts between 20 and 50 years of age, (iii) greying is progressive in nature (the total number and percentage of grey hairs increases over time), and (iv) the proportion of white hairs reaches high levels in old age, although some hairs can retain pigmentation until death, particularly among certain body regions (*Trueb and Tobin, 2010*). Additionally, our findings demonstrate that (v) age-related greying is naturally reversible in isolated hair follicles, at least temporarily and in individual HS, and may be acutely triggered by stressful life experiences, the removal of which can trigger reversal.

Aiming for the simplest model that accounts for these known features of hair greying dynamics, we found a satisfactory model with three components (*Figure 5B*): (1) an '*aging factor*' that progressively accumulates within each hair follicle, based on the fact that biological aging is more accurately modeled with the accumulation of damage, rather than a decline in stem cells or other reserves (*Kinzina et al., 2019*); (2) a biological *threshold*, beyond which hairs undergo depigmentation (i.e. greying), characterizing the transition between the dark and white states in the same HS; and (3) a '*stress factor*' that acutely but reversibly increases the aging factor during a stressful event. For modeling purposes, the accumulation of the aging factor is equivalent to the inverse of the decrease in a youth factor (e.g. loss of telomere length with age). Based on the mosaic nature of scalp HFs and our data indicating that not all hairs are in perfect synchrony, the proposed model for an entire population of hairs must also allow a variety of aging rates, as well as differential sensitivity to stress among individual hairs.

We find that the model's predicted hair population behavior (% of white HSs on a person's head over time) across the lifespan is consistent with expected normal human hair greying dynamics (*Figure 5C*). White hairs are largely absent until the onset of greying past 20 years of age then accumulate before finally reaching a plateau around 70–90% of white hairs, near 100 years. Thus, this model recapitulates the expected between-hair heterogeneity of greying within an individual, producing the common admixture of white and pigmented hairs or 'salt and pepper' phenotype in middle-age. However, some individuals also develop hairs with intermediate pigmentation states (i.e. silver/steel color), which our model does not reproduce. This represents a limitation to be addressed in future research.

We note that there are natural inter-individual differences in the rate of greying: some individuals begin greying early (onset in early 20's); some begin late (onset in 50's). A higher rate of accumulation of the aging factor (higher slope for each hair) or a lower threshold naturally accounts for earlier onset of greying. In addition, our model reveals that within a person, greater hair-to-hair heterogeneity in the rate of aging between HFs, modeled as the standard deviation of slope across hairs, also

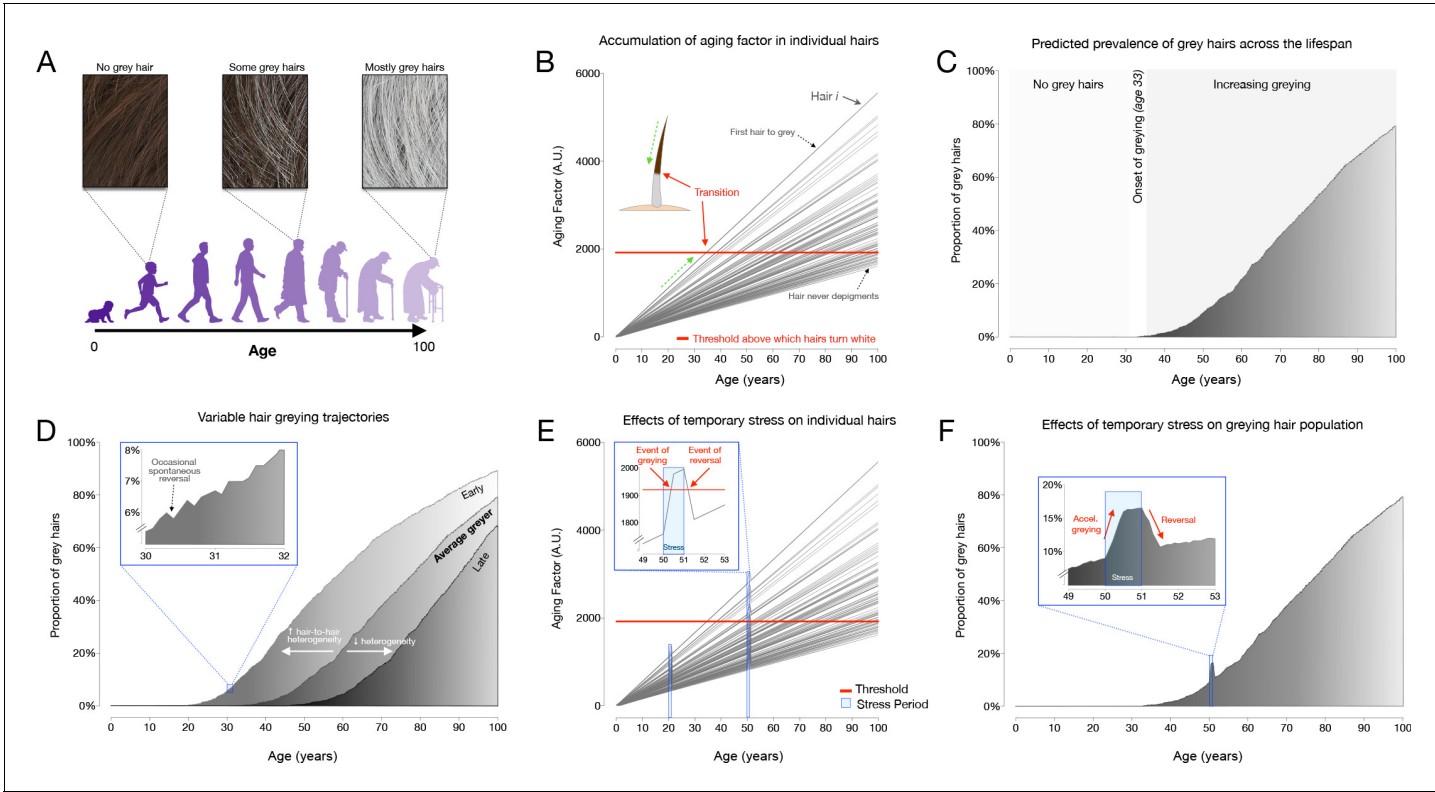

**Figure 5.** Modeling of hair greying and reversal across the human lifespan and in response to temporary stress. (**A**) Schematic overview of the average greying process across the lifespan involving the gradual loss of pigmentation, or accumulation of white hairs, mostly in the second two-thirds of life. (**B**) Depiction of individual hairs (each line is a hair, i) from a linear mixed effects model with random intercept and slopes predicting hair greying. The model assumes (i) a constant increase in a putative aging factor and (ii) a constant threshold above which hairs undergo depigmentation. All model parameters are listed in *Supplementary file 4*. (**C**) Predicted hair population behavior (n = 1000 hairs) shown as a cumulative frequency distribution of white hairs from the individual trajectories in panel (**B**). (**D**) Frequency distributions of grey hairs for individuals with early (*left*), average (*middle*), or late (*right*) hair greying phenotypes. The inset highlights a 2-year period during mid-life indicating gradual accumulation of white hairs, as well as the spontaneous repigmentation of a small fraction of white hairs (decrease in % white hairs), consistent with real-world data. (**E**) Single hair-level and (**F**) hair population-level results from the addition of two acute stress periods (each one year in duration, occurring at ages 20 and 50). The optimized model accounts for stress-induced greying in hairs whose aging factor is close to the depigmentation threshold, but not for young hairs or those far away from the threshold. Similarly, the removal of the stressor causes repigmentation of hairs when the aging factor returns below the threshold. The online version of this article includes the following source data and figure supplement(s) for figure 5:

**Figure supplement 1.** High-resolution analysis of hair pigmentation patterns.

**Figure supplement 1—source data 1.** Source data for *Figure 5—figure supplement 1*.

**Figure supplement 2.** Alternative modeling of HPP greying transitions in response to stress.

influences the onset of greying. Greater heterogeneity between HFs allows for earlier onset of greying, whereas decreasing hair-to-hair variation (i.e. lower heterogeneity) is associated with a 'youthful' later onset of greying (*Figure 5D*). Interestingly, this unpredicted result aligns with the notion that increased cell-to-cell heterogeneity is a conserved feature of aging (*Bahar et al., 2006*; *Enge et al., 2017*; *Martinez-Jimenez et al., 2017*) and that biological heterogeneity can predict all-cause mortality in humans (*Patel et al., 2010*).

## Modeling stressors produce hair greying and temporary reversal

Using parameter values that yield the average onset and rate of greying, we then simulated the influence of acute psychosocial stressors, either early in life before the onset of greying, or later once grey HSs have begun to accumulate. Similar to our data, the model also predicts transitory, or temporary reversible events of greying (see *Figure 3D*). Transitory greying events do not affect all hairs, only those that are close to the threshold at the time of stress exposure undergo greying. Hairs whose cumulative aging factors are substantially lower than threshold do not show stress-induced

greying (a 5-year-old is unlikely to get grey hairs from stress, but a 30-year-old can) (*Figure 5E–F*). Similarly, grey hairs far above threshold are not affected by periods of psychosocial stress. Thus, our model accounts for both the overall hair greying dynamics across the lifespan, and how a stressor (or its removal) may precipitate (or cause reversal of) greying in hairs whose aging factor is close to the greying threshold.

We speculate that this simulation opens an attractive possibility whereby HPP data from individuals could be used in models to formulate predictions about future greying trajectories, and to use HPPs and hair population greying behavior to track the effectiveness of behavioral and/or therapeutic interventions aimed at modifying human aging. Extending our high-resolution quantitative digitization approach to hundreds of randomly sampled dark non-transitioning hairs from different scalp regions in the same individuals, we also show that fully dark (i.e. non-greying) HSs exhibit mostly unique HPPs, but that hairs among the same scalp regions may exhibit more similar HPPs than hairs sampled from different regions (*Figure 5—figure supplement 1*; *Stenn and Paus, 2001*). This may in part be influenced by the migration of stem cells during embryogenesis to different parts of the scalp, or by other unknown factors. This preliminary extension of the HPP methodology provides a foundation for future studies. Moreover, the regional segregation of HPPs may reflect well-recognized regional differences in the rate of HS formation (*Robbins, 2012*). Thus, future models may also be able to leverage information contained within HPPs from non-greying hairs and make specific inference from hairs collected across scalp regions. Similar to how decoding temporal patterns of electroencephalography (EEG) provides information about the state of the brain, our data make it imaginable that decoding HPP analysis over time may provide information about the psychobiological state of the individual.

## Discussion

Our approach to quantify HPPs demonstrates rapid greying transitions and their natural transitory reversal within individual human hair follicles at a higher frequency and with different kinetics than had previously been appreciated. The literature generally assumes pigment production in the HFPU to be a continuous process for the entire duration of an anagen cycle, but here we document a complete switch-on/off phenomena during a single anagen cycle. The proteomic features of hair greying directly implicate multiple metabolic pathways that are both reversible in nature and sensitive to stress-related neuroendocrine factors. Therefore, this result provides a plausible biological basis for the rapid reversibility of greying and its association with psychological factors, and also supports the possibility that this process could be targeted pharmacologically.

Melanogenesis is also known to both involve and respond to oxidative stress, a byproduct of mitochondrial metabolic processes (*Balaban et al., 2005*) and driver of senescence (*Vizioli et al., 2020*). Moreover, alterations in energy metabolism are a major contributor to other disease-related aging features (*Kennedy et al., 2014*), including lifespan regulation (*Jang et al., 2018*; *Latorre-Pellicer et al., 2016*). The upregulation of specific components related to mitochondrial energy metabolism in white hairs suggests that energy metabolism regulates not only hair growth as previously demonstrated (*Flores et al., 2017*; *Mancino et al., 2020*; *Vidali et al., 2014*) but also HF greying biology. Our findings demonstrating an upregulation of the fatty acid synthesis and metabolism machinery resonate particularly strongly with recent work demonstrating that fatty acid synthesis by FASN (*Fafián-Labora et al., 2019*) and transport by CPT1a (*Seok et al., 2020*) are sufficient drivers of cell senescence, and that fatty acid metabolism regulates melanocyte aging biology (*Tang et al., 2019*). Approaches combining both high molecular and spatial resolution may be particularly informative (*Vyumvuhore et al., 2021*). Causally linking these putative metabolic changes to canonical aging and/or senescence markers in human hair shafts and among specific HF cell populations will be an important challenge for the field.

Although surprising, the reversal of hair greying is not an isolated case of 'rejuvenation' in mammals. In vivo, exposing aged mice to young blood in parabiosis experiments (*Rebo et al., 2016*; *Villeda et al., 2014*) or diluting age-related factors in old animals (*Mehdipour et al., 2020*) triggers the reversal of age-related molecular, structural and functional impairments. In human cells, quantitative biological age indicators such as telomere length (*Puterman et al., 2018*) and DNA methylation (*Fahy et al., 2019*) also exhibit temporary reversal in response to exercise and dietary interventions. Moreover, the reversibility of greying in aging human HFs demonstrated by our data is also

consistent with the observed reversibility of human skin aging in vivo when aged human skin is xeno-transplanted onto young nude mice (*Gilhar et al., 1991a*). Notably this skin 'rejuvenation' is associated with a marked increase in the number of melanocytes in human epidermis (*Gilhar et al., 1991b*), suggesting plasticity of the melanocyte compartment. Therefore, our HPP data and simulation model adds to a growing body of evidence demonstrating that human aging is not a linear, fixed biological process but may, at least in part, be halted or even temporarily reversed. Our method to map the rapid (weeks to months) and natural reversibility of human hair greying may thus provide a powerful model to explore the malleability of human aging biology within time scales substantially smaller than the entire lifespan.

A notable finding from both proteomics experiments is the bias toward *up*regulation rather than the loss of proteins in depigmented grey HS. As noted above, this may reflect the fact that hair greying is an actively regulated process within the HPFU, and that aging is not marked by a loss, but rather an increase in heterogeneity and biological complexity (*Bahar et al., 2006*; *Enge et al., 2017*; *LaRocca et al., 2020*; *Martinez-Jimenez et al., 2017*). Relative to the youthful state, quiescent and senescent cells exhibit upregulation of various secreted factors (*van Deursen, 2014*), as well as elevated metabolic activities (*Lemons et al., 2010*), rather than global downregulation of cellular activities. Moreover, similar to the macroscopic appearance of hair greying, age-related senescence markers naturally occur stochastically for DNA methylation changes across the genome (*Franzen et al., 2017*) and among cells heterogeneously scattered within tissues in mice (*Baker et al., 2016*; *Omori et al., 2020*). Our data reveal that the conserved principle of an age-related increase in molecular and cellular heterogeneity is reflected not only at the tissue level (mixture of dark and white hairs) but also in the greying hair proteome.

Moreover, our proteomics results are also in line with recent reports of keratin-associated proteins that are downregulated in white vs dark hairs (*Giesen et al., 2011*), and proteins that are upregulated with increasing age of the donor (*Plott et al., 2020*). Specifically, of a previously identified group of 50 potentially age-related, upregulated proteins in the HS (*Plott et al., 2020*), 16 were detected in our second proteomic experiment. Of these 16, 14 were similarly upregulated in depigmented white hairs relative to dark hairs from the same individuals in our dataset (*Supplementary file 2*). Further work will be required to determine if specific molecular aging processes, in specific cell types within the HF, account for the visible macroscopic instability of HFs greying on the human scalp.

Finally, in relation to psychobiological processes, the spatio-temporal resolution of the HPP approach provides investigators with an instructive new research tool that allows to link, with an unprecedented level of resolution, hair greying/reversal events with psychosocial exposures. Here, we provided proof-of-concept evidence that biobehavioral factors are linked to human hair greying dynamics. Our optical digitization approach thus extends previous attempts to extract temporal information from human hairs and illustrate the utility of HPP profiling as an instructive and sensitive psychobiology research model. Additional prospective studies with larger sample sizes are needed to confirm the robust reproducibility and generalizability of our findings. Visualizing and retrospectively quantifying the association of life exposures, stress-associated neuroendocrine factors, and HPPs may thus contribute to elucidating the mechanisms responsible for the embedding of stress and other life exposures in human biology.

## Acknowledgements

Work in the author's laboratory was supported by the Wharton Fund and NIH grants GM119793, MH119336, and AG066828 (MP). The authors are grateful to Marko Jovanonic for advice on hair proteomics, Avsar Rana, David Sulzer, Erin Seifert, and Mary Elizabeth Sutherland for valuable input at different stages of this project, and to participants who donated hairs and time for this study. The authors are also grateful to the Proteomics Shared Resource, HICCC, Columbia University Irving Medical Center (2P30 CA013696-45 Cancer Center Support Grant).

## Additional information

### Funding

| Funder | Grant reference number | Author |
|--------|------------------------|--------|
| Nathaniel Wharton Fund | | Martin Picard |
| National Institute of General Medical Sciences | GM119793 | Martin Picard |
| National Institute of Mental Health | MH119336 | Martin Picard |
| National Institute on Aging | AG066828 | Martin Picard |

The funders had no role in study design, data collection and interpretation, or the decision to submit the work for publication.

### Author contributions

Ayelet M Rosenberg, Data curation, Formal analysis, Visualization, Methodology, Writing - review and editing; Shannon Rausser, Data curation, Formal analysis, Visualization, Writing - review and editing; Junting Ren, Software, Writing - review and editing; Eugene V Mosharov, Data curation, Validation, Writing - review and editing; Gabriel Sturm, Conceptualization, Data curation, Formal analysis, Writing - review and editing; R Todd Ogden, Software, Formal analysis, Writing - review and editing, Developed the simulation model; Purvi Patel, Data curation; Rajesh Kumar Soni, Data curation, Writing - review and editing; Clay Lacefield, Formal analysis, Methodology, Writing - review and editing; Desmond J Tobin, Writing - review and editing; Ralf Paus, Writing - original draft; Martin Picard, Conceptualization, Supervision, Funding acquisition, Investigation, Writing - original draft, Project administration, Writing - review and editing

### Author ORCIDs

Ayelet M Rosenberg (iD) https://orcid.org/0000-0002-3575-5871
Desmond J Tobin (iD) http://orcid.org/0000-0003-4566-9392
Martin Picard (iD) https://orcid.org/0000-0003-2835-0478

### Ethics

Human subjects: The study was approved by New York State Psychiatric Institute (NYSPI IRB Protocol #7748). All participants provided written informed consent for their participation in this study and to the publications of data.

### Decision letter and Author response

Decision letter https://doi.org/10.7554/eLife.67437.sa1
Author response https://doi.org/10.7554/eLife.67437.sa2

## Additional files

### Supplementary files

• Supplementary file 1. Proteomic changes in greying hairs from experiment 1. Fold changes are average abundance values for white relative to dark hair shafts (HS), determined by LC-MS/MS proteomics on matched white and dark HS from female and male individuals. The last column indicates overlap with known mitochondrial proteins listed in MitoCarta 2.0 and other database annotations.

• Supplementary file 2. Proteomic changes in greying hairs from experiment 2. Fold changes are average abundance values for white relative to dark hair shafts (HS), determined by LC-MS/MS proteomics on white and dark HS from three female and three male individuals. The fourth column indicates overlap with known mitochondrial proteins listed in MitoCarta 2.0 and other database annotations. The fifth column indicates overlap with proteins that were found to be upregulated with increasing age of the donor (*Plott et al., 2020*).

• Supplementary file 3. LC-MS/MS proteomic changes in greying hairs from experiment 2. Proteins undetected in dark hairs but detected in white HS from three female and three male individuals. Because these proteins are undetected in dark hairs, the fold change is infinity. The last column indicates overlap with known mitochondrial proteins listed in MitoCarta 2.0 and other database annotations.

• Supplementary file 4. Parameters used in hair greying simulation models. Model parameters, descriptions, and values used in simulation models shown in *Figure 5*. Default values in the online simulation match those of *Figure 5E and F*.

• Transparent reporting form

### Data availability

All data generated and analyzed during this study are included in the supporting data files. Source data files have been provided for Figures 1, 2, 3, and 4, and for figure supplements (Figure 1—figure supplement 4, Figure 1—figure supplement 5, Figure 2—figure supplement 1, Figure 3—figure supplement 2, Figure 4—figure supplement 1, Figure 5—figure supplement 2). Source code for the hair simulation model is available on the App and on GitHub at https://github.com/junting-ren/hair_simulation (copy archived at https://archive.softwareheritage.org/swh:1:rev:3a19705969bfca7edc98651c1dd973ca7ae3b23d).

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
