## [Decision Letter]

Acceptance summary:

This is an interesting and informative study reporting on the molecular features of reversible hair greying in humans and the connection with psychological stress. This work will set the stage for future mechanistic studies and represents an important conceptual and methodological advance.

Decision letter after peer review:

Thank you for submitting your article "Quantitative Mapping of Human Hair Greying and Reversal in Relation to Life Stress" for consideration by *eLife*. Your article has been reviewed by 3 peer reviewers, including Matt Kaeberlein as the Senior and Reviewing Editor and Reviewer #1. The following individual involved in review of your submission has agreed to reveal their identity: Michael P Philpott (Reviewer #2).

Essential Revisions (for the authors):

1. The interpretation of the reported -omics changes remains somewhat superficial. A more in-depth discussion of the pathways found changed in the greying process would be appreciated.

2. Were any psychosocial stress related hormones such as glucocorticoids, catecholamines, growth hormone or prolactin in the grey/white hair fibers detected?

3. The data suggest changes in fatty acid metabolism with loss of pigmentation. Changes in fatty acid metabolism are associated with senescence. Did the authors detect any markers of senescence in their study?

*Reviewer #1 (Recommendations for the authors):*

This is an interesting and informative study reporting on the molecular features of reversible hair graying in humans and the connection with psychological stress. The study appears to have been very well conducted and the interpretations are generally supported by the data. While the results are primarily correlative at this stage, this work will set the stage for future more mechanistic studies and represents an important conceptual and methodological advance.

*Reviewer #2 (Recommendations for the authors):*

I have very few comments.

I may have missed it in the detailed manuscript but did the authors detect any psychosocial stress related hormones such as glucocorticoids, catecholamines, growth hormone or prolactin in the grey/white hair fibres.

Further the data suggest changes in fatty acid metabolism with loss of pigmentation. Changes in fatty acid metabolism are associated with senescence. Did the authors detect any markers of senescence in their study

*Reviewer #3 (Recommendations for the authors):*

The only weakness of the manuscript is that the interpretation of the reported omics changes remains somewhat superficial. A more in-depth discussion of the pathways found changed in the greying process would be appreciated. Of course, it would also be of interest to see a higher N in a number of the presented assays, however, the presented data appears to suffice the character of a pilot study aiming at the establishment of a new method by providing a very detailed analysis of individuals samples.

---

## [Author Response]

Essential Revisions (for the authors):1. The interpretation of the reported -omics changes remains somewhat superficial. A more in-depth discussion of the pathways found changed in the greying process would be appreciated.

Thank you for this suggestion to help strengthen the discussion of our results. We focused our discussion on the most robust and unambiguous results, which are somewhat limited given the challenge to extract and detect a fair number of protein from the resistant hair matrix. We have expanded our discussion of the greying pathways in the discussion:

(p.42), “The upregulation of specific components related to mitochondrial energy metabolism in white hairs suggests that energy metabolism regulates not only hair growth as previously demonstrated (Flores et al., 2017; Mancino et al., 2020; Vidali et al., 2014) but also HF greying biology. […] Causally linking these putative metabolic changes to canonical aging and/or senescence markers in human hair shafts and among specific HF cell populations will be an important challenge for the field.”

2. Were any psychosocial stress related hormones such as glucocorticoids, catecholamines, growth hormone or prolactin in the grey/white hair fibers detected?

Excellent question. We agree that it would be valuable to detect stress hormones in parallel with the hair pigmentation pattern and proteomic changes at the single-hair level. To our knowledge, current methods require multiple milligrams of hair for analysis (Sauve, Koren, Walsh, Tokmakejian, and Van Uum, 2007) and are only done on bulk hair material, rather than on single hairs. Therefore, it was not possible given existing technology to capture this kind of data longitudinally along single hairs.

3. The data suggest changes in fatty acid metabolism with loss of pigmentation. Changes in fatty acid metabolism are associated with senescence. Did the authors detect any markers of senescence in their study?

We did not detect any canonical markers of senescence in our proteomic results.

However, although not a direct marker of senescence, previous work has showed that HF aging is associated with a marked decline in 2 hair keratins and 7 keratin-associate proteins (KAPs) (Giesen et al., 2011). Our proteomic experiment on the single-hair from which we analyzed 6 segments similarly showed a downregulation of 4 of those aging-related keratins/KAPs (KRT33A, KRTAP4-2, KRTAP4-3, KRTAP4-4) in the white segments as compared to the preceding dark segments. We now briefly mention these findings in the manuscript, but the downregulation for some of these proteins was not very robust, as we previously mentioned in the manuscript (p.36) In addition, a recent paper compared the proteomes of hairs from the same participant collected at age 1 and 45 (44 years difference, but early in life) (Plott et al., 2020), finding 50 proteins upregulated in the older hairs. Our data comparing white to dark hairs showed that 14 of the reported upregulated proteins were also upregulated in the white hairs in our second proteomic experiment.

The revised manuscript now addresses this point in the discussion: (p.44)

“Moreover, our proteomics results are also in line with recent reports of keratin-associated proteins that are downregulated in white vs dark hairs (Giesen et al., 2011), and proteins that are upregulated with increasing age of the donor (Plott et al., 2020). […] Of these 16, 14 were similarly upregulated in depigmented white hairs relative to dark hairs from the same individuals in our dataset (**Supplementary File 2).”**

We have added a column to Supplementary file 2 to indicate which of the proteins that we detected were also found to be upregulated in the older hairs in the Plott et al. paper.

In our view, these findings should be regarded as converging but not definitive evidence, and additional studies in both hair shafts and follicles are needed, using canonical markers of senescence.

Reviewer #2 (Recommendations for the authors):I have very few comments.I may have missed it in the detailed manuscript but did the authors detect any psychosocial stress related hormones such as glucocorticoids, catecholamines, growth hormone or prolactin in the grey/white hair fibres.

Thank you for a thoughtful review. As discussed above, there are currently no available method to examine stress hormones at this resolution (see above response to Essential Revisions #2) – but this would be a terrific addition for future experiments. We have now integrated this important suggestion into the closing statement of the revised Discussion (p.44):

“Visualizing and retrospectively quantifying the association of life exposures, stress-associated neuroendocrine factors, and HPPs may thus contribute to elucidating the mechanisms responsible for the embedding of stress and other life exposures on human biology.”

Further the data suggest changes in fatty acid metabolism with loss of pigmentation. Changes in fatty acid metabolism are associated with senescence. Did the authors detect any markers of senescence in their study

Excellent point. As explained above (see above response to Essential Revisions #3), the observed changes in protein abundance related to fatty acid metabolism (FASN, CPT1a) are entirely consistent with a pro-senescence state. However, our proteomics results did not contain any canonical senescence marker. We now discuss this in detail on p.42 of the discussion.

Reviewer #3 (Recommendations for the authors):The only weakness of the manuscript is that the interpretation of the reported omics changes remains somewhat superficial. A more in-depth discussion of the pathways found changed in the greying process would be appreciated. Of course, it would also be of interest to see a higher N in a number of the presented assays, however, the presented data appears to suffice the character of a pilot study aiming at the establishment of a new method by providing a very detailed analysis of individuals samples.

Thank you for this recommendation. As described above (see above response to Essential Revisions #1), keeping with the limitation of the hair dataset, we have now expanded the discussion of our –omics data to address the reviewer’s important point. In addition, we clearly acknowledge in the revised Discussion that a higher n of investigated individuals and hair shafts, namely in the proteomics and hair pigmentation profiling, is needed to confirm robust reproducibility of our findings (p. 44):

“[…] an instructive and sensitive psychobiology research model. Additional prospective studies with larger sample sizes are needed to confirm the robust reproducibility and generalizability of our findings. […]”

References:

Fafian-Labora, J., Carpintero-Fernandez, P., Jordan, S. J. D., Shikh-Bahaei, T., Abdullah, S. M., Mahenthiran, M.,... O'Loghlen, A. (2019). FASN activity is important for the initial stages of the induction of senescence. Cell Death Dis, 10(4), 318. doi:10.1038/s41419-019-1550-0

Flores, A., Schell, J., Krall, A. S., Jelinek, D., Miranda, M., Grigorian, M.,... Lowry, W. E. (2017). Lactate dehydrogenase activity drives hair follicle stem cell activation. Nat Cell Biol, 19(9), 1017-1026. doi:10.1038/ncb3575

Giesen, M., Gruedl, S., Holtkoetter, O., Fuhrmann, G., Koerner, A., and Petersohn, D. (2011). Ageing processes influence keratin and KAP expression in human hair follicles. Exp Dermatol, 20(9), 759-761. doi:10.1111/j.1600-0625.2011.01301.x

Mancino, G., Sibilio, A., Luongo, C., Di Cicco, E., Miro, C., Cicatiello, A. G.,... Dentice, M. (2020). The Thyroid Hormone Inactivator Enzyme, Type 3 Deiodinase, Is Essential for Coordination of Keratinocyte Growth and Differentiation. Thyroid. doi:10.1089/thy.2019.0557

Plott, T. J., Karim, N., Durbin-Johnson, B. P., Swift, D. P., Scott Youngquist, R., Salemi, M.,... Rice, R. H. (2020). Age-Related Changes in Hair Shaft Protein Profiling and Genetically Variant Peptides. Forensic Sci Int Genet, 47, 102309. doi:10.1016/j.fsigen.2020.102309

Sauve, B., Koren, G., Walsh, G., Tokmakejian, S., and Van Uum, S. H. (2007). Measurement of cortisol in human hair as a biomarker of systemic exposure. Clin Invest Med, 30(5), E183-191. Retrieved from https://www.ncbi.nlm.nih.gov/pubmed/17892760

Seok, J., Jung, H. S., Park, S., Lee, J. O., Kim, C. J., and Kim, G. J. (2020). Alteration of fatty acid oxidation by increased CPT1A on replicative senescence of placenta-derived mesenchymal stem cells. Stem Cell Res Ther, 11(1), 1. doi:10.1186/s13287-019-1471-y

Tang, L., Li, J., Fu, W., Wu, W., and Xu, J. (2019). Suppression of FADS1 induces ROS generation, cell cycle arrest, and apoptosis in melanocytes: implications for vitiligo. Aging (Albany NY), 11(24), 11829-11843. doi:10.18632/aging.102452

Vidali, S., Knuever, J., Lerchner, J., Giesen, M., Biro, T., Klinger, M.,... Paus, R. (2014). Hypothalamic-pituitary-thyroid axis hormones stimulate mitochondrial function and biogenesis in human hair follicles. J Invest Dermatol, 134(1), 33-42. doi:10.1038/jid.2013.286

Vyumvuhore, R., Verzeaux, L., Gilardeau, S., Bordes, S., Aymard, E., Manfait, M., and Closs, B. (2021). Investigation of the molecular signature of greying hair shafts. Int J Cosmet Sci. doi:10.1111/ics.12700